# A Competitive-Cooperative Actor-critic Framework for Reinforcement Learning

## Abstract

In the field of Deep reinforcement learning (DRL), enhancing exploration capabilities and improving the accuracy of Q-value estimation remain two major challenges. Recently, double-actor DRL methods have emerged as a promising class of DRL approaches, achieving substantial advancements in both exploration and Q-value estimation. However, existing double-actor DRL methods feature actors that operate independently in exploring the environment, lacking mutual learning and collaboration, which leads to suboptimal policies. To address this challenge, this work proposes a generic solution that can be seamlessly integrated into existing double-actor DRL methods by promoting mutual learning among the actors to develop improved policies. Specifically, we calculate the difference in actions output by the actors and minimize this difference as a loss during training to facilitate mutual imitation among the actors. Simultaneously, we also minimize the differences in Q-values output by the various critics as part of the loss, thereby avoiding significant discrepancies in value estimation for the imitated actions. We present two specific implementations of our method and extend these implementations beyond double-actor DRL methods to other DRL approaches to encourage broader adoption. Experimental results demonstrate that our method effectively enhances four state-of-the-art (SOTA) double-actor DRL methods and five other types of SOTA DRL methods across four MuJoCo tasks, as measured by return.

## 1 Introduction

DRL has emerged as a leading machine learning approach due to its ability to autonomously learn policies for task completion, with applications across various domains such as recommendation systems Afsar et al. (2022), autonomous driving Kiran et al. (2021), and robot planning and control Singh et al. (2022). Advancing DRL requires overcoming two key challenges: improving exploration capabilities and enhancing Q-value estimation accuracy. Improved exploration enables DRL agents to uncover more optimal policies, while refined Q-value estimation provides a more accurate assessment of action values, facilitating the discovery of superior policies. The recent introduction of double-actor DRL methods represents a significant advancement in this field, effectively addressing both challenges and allowing agents to learn more optimal policies, thereby establishing themselves as highly promising methods in DRL.

The double-actor DRL methods incorporate two actors and two critics, with each actor paired with a separate critic. These actor-critic pairs are trained sequentially. This structure allows both actors to interact with the environment concurrently, while a real-time action selection strategy determines the best action from the two actors, fostering competition between them. This approach offers two primary benefits: **First**, the parallel operation of two actors enhances exploration capabilities, as demonstrated in studies such as Li & Liu (2022), Xie & Li (2023), Nguyen & Luong (2023), Xiao et al. (2023), and Li et al. (2023b). **Second**, the double-actor methods improve Q-value estimation accuracy by employing regularized critics, as evidenced by Pan et al. (2020), Pan et al. (2021), Guo et al. (2024), Lyu et al. (2022), Lyu et al. (2023), and Li et al. (2022b). Beyond algorithmic and theoretical contributions, double-actor methods have been applied to real-world scenarios, including energy management Wang et al. (2022)Hu et al. (2024), federated learning Zhou et al. (2024)Chen et al. (2023a), and human motion simulation Li et al. (2022a).

However, current double-actor DRL methods encounter a significant limitation that hinders their ability to develop more effective policies. Specifically, these methods involve two actors that independently explore and learn from the environment without any cooperation. This absence of collaboration prevents the actors from leveraging each other's experiences, thereby limiting their effectiveness in developing better policies. For example, in scenarios where double-actor DRL methods are employed to control a humanoid robot navigating a cluttered environment to avoid collisions Li et al. (2022a), the two actors may pursue different path-planning strategies. However, their lack of collaboration prevents them from leveraging each other's policies, resulting in suboptimal policies and inefficient task completion.

Motivated by the observations above, this work aims to develop a generic framework that can be seamlessly integrated with existing double-actor DRL methods to promote cooperation among actor-critic pairs, facilitating the learning of better policies. Specifically, we compute the differences between actions produced by each actor and the discrepancies between Q-values provided by each critic. These differences are combined to formulate a collaborative loss, which is incorporated into the loss function of each actor-critic pair and minimized during training. By reducing the differences between actions, actors can better mimic each other's policies and derive mutual benefits. This imitation process reduces the differences in the actions taken by different actors, which in turn motivates us to minimize Q-value differences and avoid significant evaluation discrepancies among critics for the imitated actions. We introduce two methods for calculating the collaborative loss. The first method directly enables actors to imitate each other's actions and reduces the discrepancies between Q-values produced by different critics. The second method employs a value function to assess the advantages of actions taken by each actor, selectively imitating those with higher advantages to avoid replicating lower-quality actions Nikishin et al. (2022) during the trial-and-error process.

Moreover, since current double-actor DRL methods utilize two critics, and the DRL community has demonstrated that employing more critics—such as triple-critic methods Saglam et al. (2024b)—can lead to more robust DRL policies, multi-critic DRL methods are gradually emerging as a promising approach. Therefore, following the framework of double-actor DRL, we extend our approach to multi-critic DRL. Specifically, we assign an actor to each critic in the multi-critic DRL method, forming $n$ mutually competitive and imitative actor-critic pairs, where $n$ denotes the number of critics in the baseline multi-critic DRL method. This enhancement improves the exploration capabilities of multi-critic DRL methods, facilitating the learning of better policies.

In summary, our contributions are as follows: (1) We develop a generic competitive-cooperative framework that advances existing double-actor methods. Additionally, we extend this approach to a multi-critic DRL framework. (2) Our framework fosters both competition and imitation among multiple actor-critic pairs, facilitating the development of superior policies. (3) We present two specific implementations of the proposed framework and demonstrate through extensive experiments that both implementations significantly improve the performance of nine SOTA DRL methods across four MuJoCo tasks in terms of return.

## 2 RELATED WORK

**DRL**: The primary goal of DRL is to identify an optimal policy that maximizes cumulative discounted rewards. DRL was formally established in 2015 with Minh et al.'s Mnih et al. (2015) introduction of deep neural networks for high-dimensional policy learning, which has since facilitated its successful application across various tasks. Current mainstream DRL algorithms typically utilize an architecture based on one actor and two critics. Notable challenges in DRL include exploration versus exploitation and inaccurate Q-value estimation. Various approaches have been proposed to address these issues, such as intrinsic motivation methods Forestier et al. (2022)Colas et al. (2022) to enhance exploration and triple-critic methods Saglam et al. (2024b)Yin et al. (2024)Saglam et al. (2021) to improve Q-value accuracy. However, these methods often address only one challenge and are difficult to integrate with each other.

**Double-actor DRL**: Double-actor DRL methods have emerged as highly promising approaches, demonstrating exceptional performance in addressing the aforementioned challenges. They notably enhance Q-value estimation accuracy, as evidenced by Pan et al. Pan et al. (2020) with their softmax-regularized method, which was later adapted for multi-agent DRL Pan et al. (2021). Subsequent advancements by researchers such as Guo et al. Guo et al. (2024), Lyu et al. Lyu et al. (2022;

2023), and Li et al. Li et al. (2022b) have further refined Q-value estimation. These methods also improve exploration, as demonstrated by Li et al. Li & Liu (2022) with their Master-Slave policy, which complements similar approaches by Xie et al. Xie & Li (2023), Nguyen et al. Nguyen & Luong (2023), Xiao et al. Xiao et al. (2023), and Li et al. Li et al. (2023b). Beyond theoretical research, double-actor DRL methods have been widely applied in practical areas such as federated learning Zhou et al. (2024), energy management Wang et al. (2022), and resource allocation Chen et al. (2023a). Despite these advancements, double-actor DRL methods utilize two independently exploring actors, which lacks mutual cooperation and learning. This limitation prevents the two actors from benefiting from each other's policies, resulting in suboptimal policies.

**Multi-actor DRL**: Current research has shown that utilizing multiple actors is an effective strategy for enhancing the exploration capabilities and sample efficiency of DRL. One approach involves establishing multiple parallel DRL learners that collaborate and improve sample efficiency through experience sharing, as exemplified in evolutionary DRL Zhu et al. (2023), distributed DRL Chen et al. (2023b), and federated DRL Qi et al. (2021). Our method can be integrated as a single learner within a distributed environment, thereby allowing for integration within these types of methods. Another approach Li et al. (2023a)Li et al. (2024)Sheng Fan & Jiang (2020)Lin Li (2023) employs several multi-styled actors to promote diverse exploration, further enhancing the exploration capabilities of DRL. Our method can also be incorporated into this approach by assigning an additional actor to each critic. In summary, while our method differs mechanistically from existing multi-actor approaches, it is complementary and can be effectively combined with them. Additionally, unlike multi-agent DRL methods, our approach is designed to address single-agent learning tasks.

## 3 PRELIMINARIES

This section details the frameworks of double-actor DRL and triple-critic DRL, two promising DRL methods beyond regular DRL (one actor, two critics). These serve as the research background. We then present the problem formulation to elucidate the objective function.

### 3.1 EXISTING DRL METHODS

**(1) Double-actor DRL**: Fig. 1 (a) depicts the double-actor architecture, which differs from regular DRL in two primary aspects. First, in action selection, the double-actor DRL approach pairs two actors with two critics, where each actor generates an action. The paired critics then evaluate these actions' Q-values, and the action with the highest Q-value is chosen for interaction with the environment. Second, in model updates, we sample a mini-batch from the replay buffer and update each actor-critic pair independently. As in regular DRL, we calculate the Temporal Difference (TD) error by comparing the estimated target Q-value with the critic's current Q-value and use the TD error to update the critic. Subsequently, the updated critic network is used to update the paired actor.

**(2) Multi-critic DRL**: Multi-critic DRL is also a commonly used approach, with the triple-critic DRL being a representative method. Fig. 1 (d) illustrates the architecture of triple-critic DRL. This method employs one actor to generate actions that interact with the environment, along with three critics, each producing distinct Q-values. These Q-values are then combined with their corresponding target critics to compute a target Q-value for calculating the TD error, which is used to update each critic. Prior studies Saglam et al. (2024b)Yin et al. (2024)Saglam et al. (2021) have theoretically and empirically demonstrated that this approach significantly enhances the accuracy of Q-value estimation, thereby enabling DRL to develop more effective policies.

### 3.2 PROBLEM FORMULATION

In DRL training, the framework is structured as a Markov decision process (MDP), denoted by $\mathcal{M} = \langle \mathcal{S}, \mathcal{A}, r, \mathbb{P}, \gamma \rangle$. Here, $\mathcal{S}$ represents the state space, $\mathcal{A}$ denotes the action space, $r$ is the reward function, $\mathbb{P}$ is the transition matrix, and $\gamma$ is the discount factor. At each time step $t$, the agent interacts with the environment by selecting an action $a_t$ from $\mathcal{A}$ based on the current state $s_t \in \mathcal{S}$. This action is determined by the policy $\pi : \mathcal{S} \to \mathcal{A}$, and results in a reward $r_t$. The Q function, $Q_\pi(s, a)$, which estimates the expected cumulative reward of following policy $\pi$ starting from state $s$ and taking action $a$, is defined as $Q_\pi(s, a) = \mathbb{E}_\pi \left[ \sum_{i=t}^{T} \gamma^{i-t} r_i \mid s_t = s, a_t = a \right]$.

A common approach in DRL is the actor-critic method, where the actor $\phi$ determines the policy $\pi(s; \phi)$, and the critic $\theta$ estimates the Q function $Q(s, a; \theta)$. In double-actor DRL methods, the agent utilizes two separate actor networks $\phi = (\phi_1, \phi_2)$ to produce different actions, $a_t^1$ and $a_t^2$, from the same state $s_t$. These actions are then assessed by two distinct critics, $\theta^{Q_1}$ and $\theta^{Q_2}$, which evaluate the actions as $Q_1(s_t, a_t^1)$ and $Q_2(s_t, a_t^2)$, respectively. The agent chooses $a_t^1$ if its Q-value $Q_1(s_t, a_t^1)$ surpasses $Q_2(s_t, a_t^2)$; otherwise, $a_t^2$ is selected. The objective of double-actor DRL methods is to learn an optimal policy $\pi(s; \phi)$ that maximizes the long-term cumulative reward, formalized as $J(\pi(\phi)) = \max_\phi \mathbb{E}_{\pi(\phi)} \left[ \sum_{i=0}^T \gamma^{i-t} r_i \mid s_0, a_0 \right]$.

However, existing double-actor DRL methods face a limitation. The independent exploration by the two actors lacks cooperation, as each actor learns only from its own experiences rather than collaborating. This results in suboptimal policies and lower long-term cumulative rewards $J(\pi(\phi))$.

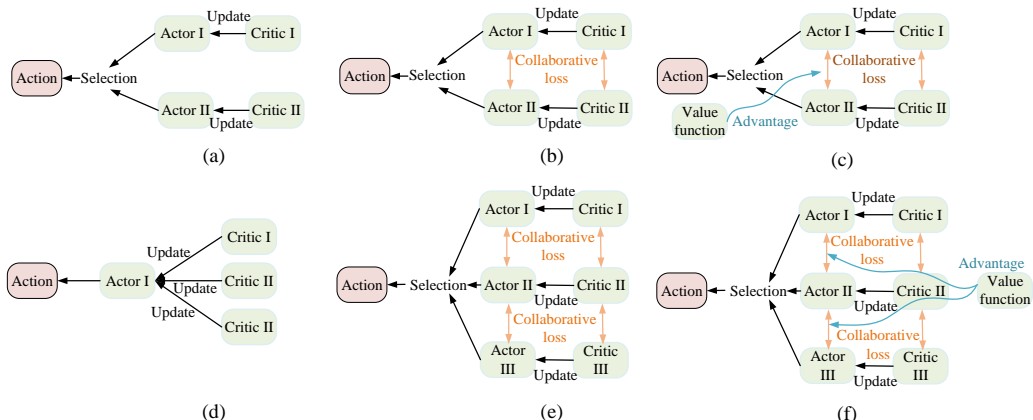

Figure 1: Framework of existing DRL methods and integration of our approach. Subfigures (a) and (d) illustrate the existing double-actor DRL and multi-critic DRL methods, with triple-critic as an example. Subfigures (b) and (e) demonstrate our first implementation method for calculating the collaborative loss applied to these approaches, while subfigures (c) and (f) present our second implementation method.

## 4 METHODOLOGY

This section first outlines the framework of our method and then details the model training and action selection strategy. Finally, we present an analysis of both the temporal and spatial complexity, along with a theoretical analysis of our method. The detailed training process and pseudocode implementation are provided in the appendix.

### 4.1 THE FRAMEWORK OF OUR APPROACH

The framework for our method is illustrated in Fig. 1 (b) (c) (e) (f). Existing double-actor DRL methods involve each actor-critic pair exploring independently, without interaction between them (see Fig. 1 (a)). In contrast, our framework promotes mutual imitation among actor-critic pairs. Specifically, we introduce a collaborative loss that consists of two components: the first component minimizes the discrepancy in actions between actors from different pairs, facilitating mutual imitation of actions; the second component minimizes the variation in Q-values among critics, thereby reducing significant differences in the value estimation for the imitated actions.

We present two distinct implementations for calculating the collaborative loss. **The first implementation** involves directly computing the differences in both actor and critic outputs to form the collaborative loss, which is minimized throughout training (see Fig. 1 (b) (e)). **The second implementation** utilizes the value function to calculate the advantage of actions produced by each actor and selectively imitates actions with higher advantages, thereby avoiding the imitation of lower-quality actions during the trial-and-error process (see Fig. 1 (c) (f)). Our two implementations are generic and

can be seamlessly integrated into existing double-actor DRL methods as well as various other types of DRL methods. Fig. 1 (b) (c) demonstrate the integration of our implementations into double-actor DRL, while Fig. 1 (e) (f) illustrate the integration into multi-critic DRL.

## 4.2 MODEL TRAINING

This section details the training process for our method. Our approach involves pairing each actor with a critic and updating each actor-critic pair in a consistent manner. Specifically, we sample a mini-batch, denoted as $\mathcal{D}_0$, from the replay buffer $\mathcal{D}$. Since our method can be integrated with existing DRL approaches, the sampling procedure follows the baseline DRL method. This mini-batch is then used to sequentially update each actor-critic pair, denoted as $(\phi_i, \theta^i)$. Let $n$ represent the number of actor-critic pairs. For instance, when integrating our method with triple-critic DRL Saglam et al. (2024b), $n$ is set to 3. The update process for each actor-critic pair is described below:

**(1) Calculating target Q-value**: To update each actor-critic pair, we begin by computing the target Q-value from the sampled mini-batch $\mathcal{D}_0$. The target Q-value is given by: $\hat{Q}(s_t, a_t) \leftarrow r_t + \gamma \min_{i=1,2} \mathcal{T}Q_{\theta^{i\prime}}(s_{t+1}, \tilde{a})$, where $\tilde{a} \leftarrow \pi_{\phi\prime}(s_{t+1}) + \epsilon$ with $\epsilon \sim \mathcal{N}(0, 0.2)$. Here, $\phi\prime$ represents the target actor, while $\theta^{i\prime}$ denotes the target critic. The term $\mathcal{T}Q_{\theta^{i\prime}}(s_{t+1}, \tilde{a})$ represents the next Q-value, which can be calculated in various ways depending on the double-actor DRL method employed. For example, Pan et al. Pan et al. (2020) enhance the accuracy of Q-value estimation through a softmax function applied to the next Q-value. When integrating our method with various double-actor DRL methods, it is essential to use their specific methods for computing the next Q-value. After calculating the target Q-value, we sequentially update each actor-critic pair.

**(2) Updating critic in each actor-critic pair**: Let $\alpha$ denote the learning rate. Initially, we update the critic network for each actor-critic pair (such as the $i^{th}$ critic $\theta^i$) by employing the target Q-value $\hat{Q}(s_t, a_t)$, as detailed below: $\theta^i \leftarrow \theta^i + \alpha \nabla_{\theta^i} \frac{1}{|\mathcal{D}_0|} \sum_{(s_t, a_t) \in \mathcal{D}_0} \left( \hat{Q}(s_t, a_t) - Q_{\theta^i}(s_t, a_t) \right)^2$.

**(3) Updating actor in each actor-critic pair**: After updating the critic $\theta^i$, we then update the corresponding actor (e.g., the $i$-th actor $\phi_i$).

**(3.1) Compute the collaborative loss $L_i$:** For the $i$-th actor, we first calculate the collaborative loss $L_i$ to encourage imitation between different actors (e.g., $\phi_k(s_t)$ and $\phi_i(s_t)$) while avoiding excessive discrepancies in their corresponding Q-values (e.g., $Q_k(s_t, \phi_k(s_t))$ and $Q_i(s_t, \phi_i(s_t))$). We develop two distinct methods to compute $L_i$, representing two different implementations of our approach.

**First Approach**: The first approach directly uses the difference between $\phi_k(s_t)$ and $\phi_i(s_t)$, and the difference between $Q_k(s_t, \phi_k(s_t))$ and $Q_i(s_t, \phi_i(s_t))$ as the collaborative loss $L_i$, aiming to minimize the discrepancy between them, as follows:

$$L_i = \sum_{k=1 \ k \neq i}^{n} \left( \frac{1}{|\mathcal{D}_0|} \sum_{i=1}^{|\mathcal{D}_0|} \text{Fs}(Q_k(s_t, \phi_k(s_t)) - Q_i(s_t, \phi_i(s_t)) \cdot (\phi_k(s_t) - \phi_i(s_t))^2 \right) \quad (1)$$

where $\text{Fs}(x) = \frac{1}{2}(\text{sgn}(x) + 1)$ is used to map the difference between Q-values to a range of $[0, 1]$ to prevent it from becoming too large.

**Second Approach**: The second approach does not compute the difference between the actions $\phi_k(s_t)$ and $\phi_i(s_t)$ directly as the loss $L_i$. Instead, it estimates the advantage functions for each action to determine whether $\phi_i(s_t)$ should mimic $\phi_k(s_t)$, as the advantage function evaluates how actions $\phi_k(s_t)$ and $\phi_i(s_t)$ perform relative to the average, with a higher advantage indicating a potentially higher value. Specifically, if the advantage of action $\phi_k(s_t)$ is greater than the advantage of action $\phi_i(s_t)$ of the actor $i$ being updated, then $\phi_i(s_t)$ should mimic $\phi_k(s_t)$, thus minimizing the difference between them. Conversely, if the advantage of $\phi_k(s_t)$ is less than that of $\phi_i(s_t)$, there is no need to mimic the action $\phi_k(s_t)$.

Let $\theta^v$ be the network for outputting the value function $V(s)$. The advantage functions for two actions are given by $A(s, \phi_i(s_t))$ and $A(s, \phi_k(s_t))$, where: $A(s, \phi_i(s_t)) = Q_i(s_t, \phi_i(s_t)) - V(s)$ and $A(s, \phi_k(s_t)) = Q_i(s_t, \phi_k(s_t)) - V(s)$. Next, we compare the magnitudes of these advantage functions as follows:

$$\hat{a} = \phi_i(s_t) * \mathbf{1}[A(s, \phi_i(s_t)) >= A(s, \phi_k(s_t))] + \phi_k(s_t) * \mathbf{1}[A(s, \phi_k(s_t)) >= A(s, \phi_i(s_t))] \quad (2)$$

where $\hat{a}$ represents the action that $\phi_i(s_t)$ needs to mimic. The function $\mathbf{1}[x, y]$ is an indicator function that equals 1 if $x > y$ and 0 otherwise. Subsequently, similar to the first method, we calculate the collaborative loss $L_i$, given by:

$$L_i = \sum_{\substack{k=1 \ k \neq i}}^{n} \left( \frac{1}{|\mathcal{D}_0|} \sum_{i=1}^{|\mathcal{D}_0|} \text{Fs}(Q_k\left(s_t, \phi_k(s_t)\right) - Q_i\left(s_t, \phi_i(s_t)\right) \cdot (\hat{a} - \phi_i(s_t))^2 \right) \quad (3)$$

For updating the value function network $\theta^v$, we adopt the method used in existing studies Yang et al. (2023)Liu et al. (2024). Specifically, we start by calculating the next Q-value $\tilde{Q}(s_t, a_t)$ with the current critics: $\tilde{Q}(s_t, a_t) \leftarrow r_t + \gamma \min_{i=1,2} \mathcal{T}Q_{\theta^i}(s_{t+1}, \tilde{a})$ where $\tilde{a} \leftarrow \pi_{\phi'}(s_{t+1}) + \epsilon$ and $\epsilon \sim \mathcal{N}(0, 0.2)$. Here, $\theta^i$ denotes the $i$-th current critic. The target value $\hat{V}(s)$ is then computed as: $\hat{V}(s) = \min\left(\tilde{Q}(s_t, a_t), \hat{Q}(s_t, a_t)\right)$. The value function network $\theta^v$ is updated by minimizing the difference between the target value $\hat{V}(s)$ and the current value $V(s)$, as follows: $\mathcal{L}_V(\theta^v) = \mathbb{E}_{(s,a) \sim \mathcal{D}_0}\left[\left(\hat{V}(s) - V(s)\right)^2\right]$.

**(3.2) Computing the gradient of the actor**: After calculating the collaborative loss $L_i$, we update the $i^{th}$ actor $\phi_i$ using $Q_{\theta^i}(s_t, \pi_{\phi_i}(s_t))$, $L_i$, and $\pi_{\phi_i}(s_t)$ as follows:

$$\nabla_{\phi_i} J(\phi_i) = \frac{1}{|\mathcal{D}_0|} \sum_{(s_t, a_t) \in \mathcal{D}_0} \nabla_{\pi_{\phi_i}(s_t)} Q_{\theta^i}(s_t, \pi_{\phi_i}(s_t)) \nabla_{\phi_i} \pi_{\phi_i}(s_t) + \nabla_{\pi_{\phi_i}(s_t)} L_i \quad (4)$$

### 4.3 ACTION SELECTION

In our method, which involves multiple actors, each actor generates an action, but ultimately, only one action is selected to interact with the environment. Therefore, at each step, we estimate the $Q$-value of each action and choose the action with the highest $Q$-value for interaction with the environment. This is because the Q-value represents the expected cumulative reward of that action, which corresponds to the value of the action. Consequently, actions with higher Q-values are more likely to guide the DRL agent toward achieving a higher return. Specifically, we select the action corresponding to the maximum Q-value, meaning that the selected action $a_t$ is determined by $a_t = \max\left(Q_{\theta^1}(s_t, \pi_{\phi_1(s_t)}), Q_{\theta^2}(s_t, \pi_{\phi_2(s_t)}), ..., Q_{\theta^n}(s_t, \pi_{\phi_n(s_t)})\right)$.

For example, in a two-actor DRL algorithm, the agent chooses $a_t^1$ if $Q_{\theta^1}(s_t, a_t^1)$ exceeds $Q_{\theta^2}(s_t, a_t^2)$; otherwise, it opts for $a_t^2$. However, some double-actor DRL algorithms, such as those by Li et al. Li et al. (2022b), use unique methods for action selection. Therefore, when integrating our method with these DRL algorithms, their specific approach should be applied to determine an action based on $\left(Q_{\theta^1}(s_t, \pi_{\phi_1(s_t)}), Q_{\theta^2}(s_t, \pi_{\phi_2(s_t)}), \ldots, Q_{\theta^n}(s_t, \pi_{\phi_n(s_t)})\right)$.

### 4.4 COMPLEXITY ANALYSIS

**(1) First Implementation**: The **time complexity** is as follows: For each actor-critic pair, this implementation requires $n - 1$ forward computations for both the critic and the actor across all $|\mathcal{D}_0|$ samples. Given that the complexity of the forward computation scales linearly with the length of the input sequences, where $X$ denotes the length of the actor's input sequence and $Y$ denotes the length of the critic's input sequence, the time complexity is $O(n(n-1)|\mathcal{D}_0|(X + Y))$. The **space complexity** remains unchanged compared to the existing double-actor DRL, as no additional storage is required. **(2) Second Implementation**: The **time complexity** is as follows: This implementation requires an additional forward computation for a value network. Moreover, calculating and comparing the advantage value for each action involves simple subtraction and minimal numerical comparisons, which are computationally negligible. Let $Z$ represent the length of the input sequence for the value network. Thus, the time complexity of this implementation is $O(n(n-1)|\mathcal{D}_0|(X + Y + Z))$. The

**space complexity** increases due to the parameters of the value network, which includes a five-layer Multilayer Perceptron.

The complexity analysis above shows that both implementations of our framework do not introduce excessive complexity, thereby ensuring the effectiveness and broad applicability of our approach.

## 4.5 THEORETICAL ANALYSIS

This section presents a theoretical analysis explaining why our method improves upon existing double-actor DRL approaches. A detailed derivation is included in the appendix, where the proof of Theorem 1 is based on Assumption 1 and Lemmas 1 through 5.

**Theorem 1**: Let $n$ represent the number of actor-critic pairs in our method, and let $\pi^k(s)$ denote the policy of the $k$-th actor. The difference between $J(\pi^i(s))$, which indicates the performance of the $i$-th actor's policy $\pi^i(s)$, and the optimal policy associated with this policy, $J(\pi_i^*(s))$, can be bounded as follows:

$$|J(\pi_i^*(s)) - J(\pi^i(s))| \leq$$

$$\frac{R_{\max}K_c}{1-\gamma}\left(\max_{s\in\mathcal{S}}\|\pi_i^*(s) - \pi^1(s)\| + \sum_{k=2 \ k\neq i}^{n}\left(\max_{s\in\mathcal{S}}\|\pi^{k-1}(s) - \pi^k(s)\|\right) + \max_{s\in\mathcal{S}}\|\pi^n(s) - \pi^i(s)\|\right)$$

(5)

where $\frac{R_{\max}K_c}{1-\gamma}$ is a positive constant that remains fixed for any policy.

**Discussion**: The above derivation demonstrates that facilitating mutual imitation among the policies of the actor-critic pairs can reduce $\sum_{k=2 \ k\neq i}^{n}\left(\max_{s\in\mathcal{S}}\|\pi^{k-1}(s) - \pi^k(s)\|\right) + \max_{s\in\mathcal{S}}\|\pi^n(s) - \pi^i(s)\|$, thereby lowering the upper bound on the difference between the policy $J(\pi^i(s))$ and the optimal policy $J(\pi_i^*(s))$. Consequently, the performance $J$ of $\pi^i(s)$ has the potential to more closely align with that of the optimal policy $\pi_i^*(s)$. We apply the same analysis, as outlined in Theorem 1, to each actor in the remaining actor-critic pairs within our method. This analysis suggests that each actor's policy will potentially converge more closely to its optimal policy. Therefore, by facilitating mutual imitation between actors and critics, our approach enhances the policy performance of existing double-actor DRL methods.

## 5 EXPERIMENT AND ANALYSIS

This section details the experimental setup, results, and analysis. The software used includes Torch 1.2.0, Gym 0.16.0, and Mujoco-py 1.50.0.1, while the hardware configuration comprises an Intel Core i7-9700 CPU, 64GB RAM, and an NVIDIA RTX 2080 GPU. To ensure alignment with SOTA DRL research, each method was evaluated across 5 different random seeds. Implementation details, as well as the detailed training curves in terms of episode return, are presented in the appendix.

## 5.1 EXPERIMENT PREPARATION

**(1) Baselines**. Nine SOTA DRL methods are employed as baselines. Among these, four are SOTA double-actor DRL methods: **Conservative Advantage Learning (CAL)** Li et al. (2022b), **Double-Actors Regularized Critics (DARC)** Lyu et al. (2022), **Softmax Deep Double Deterministic Policy Gradients (SD3)** Pan et al. (2020), and **Generalized-activated Deep Double Deterministic Policy Gradients (GD3)** Lyu et al. (2023). The remaining five are three SOTA multi-critic DRL methods: **Stochastic Weighted DRL (SW)** Saglam et al. (2024b), **Multi-Critic DRL (MC)** Yin et al. (2024), and **N-Step Method (N-Step)** Zhang et al. (2024), as well as two SOTA regular DRL methods: **One Step Q-learning (OSQ)** Saglam et al. (2024a), **Policy Correction (PC)** Xu et al. (2024).

**(2) Benchmarks**. To assess the performance of the methods, evaluation was conducted across four widely adopted MuJoCo environments - HalfCheetah-v2, Hopper-v2, Walker2d-v2, and Ant-v2, in line with the prevailing research practices in the field.

**(3) Metrics**. Our evaluation leverages two key performance metrics: **the episode return** (presented in the appendix) and **the final return** (presented in the main text). Specifically, **the episode return**

Table 1: Performance comparison of different methods (Mean ± Standard deviation). Here, Our SW (I) and Our SW (II) denote the first and second implementations of our approach integrated into SW, respectively. The format for other methods follows the same convention. Bold results indicate improvements over the baseline DRL algorithms.

| | Verification on double-actor DRL | | | |
|---|---|---|---|---|
| Methods | HalfCheetah | Hopper | Walker2d | Ant |
| CAL | $10917.53 \pm 1572.43$ | $2851.58 \pm 765.56$ | $4907.03 \pm 412.98$ | $1750.45 \pm 1332.22$ |
| **Our CAL (I)** | **$11689.88 \pm 832.88$** | **$3529.85 \pm 260.04$** | $4821.53 \pm 502.77$ | **$4221.99 \pm 367.92$** |
| **Our CAL (II)** | **$11794.23 \pm 736.08$** | **$3387.95 \pm 317.24$** | **$5167.02 \pm 586.97$** | **$3655.29 \pm 801.47$** |
| GD3 | $10129.37 \pm 919.5$ | $3393.59 \pm 380.03$ | $4587.67 \pm 382.88$ | $2440.6 \pm 1328.12$ |
| **Our GD3 (I)** | **$11035.36 \pm 780.56$** | **$3492.58 \pm 286.27$** | **$4755.85 \pm 438.69$** | **$2975.79 \pm 1026.03$** |
| **Our GD3 (II)** | **$11187.56 \pm 324.69$** | **$3715.83 \pm 101.42$** | **$4824.78 \pm 521.37$** | $414.06 \pm 639.46$ |
| DARC | $11407.27 \pm 345.22$ | $3430.68 \pm 304.6$ | $4445.27 \pm 582.1$ | $2609.13 \pm 2195.72$ |
| **Our DARC (I)** | $10532.1 \pm 1591.81$ | $3285.55 \pm 379.85$ | **$4736.18 \pm 804.62$** | **$3336.28 \pm 945.68$** |
| **Our DARC (II)** | **$11503.32 \pm 323.28$** | **$3605.08 \pm 134.12$** | **$5183.03 \pm 590.83$** | **$3896.48 \pm 913.17$** |
| SD3 | $10830.78 \pm 597.55$ | $2882.37 \pm 886.78$ | $4052.61 \pm 339.26$ | $1176.52 \pm 1247.96$ |
| **Our SD3 (I)** | $10607.72 \pm 1078.74$ | **$3428.79 \pm 350.96$** | **$4336.35 \pm 541.24$** | $362.71 \pm 273.28$ |
| **Our SD3 (II)** | **$10831.12 \pm 200.83$** | **$3442.23 \pm 240.14$** | **$4094.11 \pm 440.12$** | **$2756.46 \pm 764.43$** |
| | Verification on multi-critic DRL and regular DRL | | | |
| Methods | HalfCheetah | Hopper | Walker2d | Ant |
| OSQ | $11682.6 \pm 305.21$ | $3255.49 \pm 913.34$ | $4385.93 \pm 642.3$ | $4286.98 \pm 1224.85$ |
| **Our OSQ (I)** | $11112.59 \pm 489.87$ | **$3491.6 \pm 175.18$** | **$5095.08 \pm 355.33$** | **$5649.59 \pm 308.16$** |
| **Our OSQ (II)** | $11223.74 \pm 630.23$ | **$3333.12 \pm 526.59$** | **$5019.34 \pm 683.04$** | **$5556.08 \pm 287.76$** |
| PC | $11565.49 \pm 318.99$ | $2564.56 \pm 800.23$ | $4715.21 \pm 432.07$ | $3842.94 \pm 1010.5$ |
| **Our PC (I)** | $10842.41 \pm 311.09$ | **$3502.57 \pm 137.45$** | **$4835.24 \pm 518.26$** | **$5080.99 \pm 918.24$** |
| **Our PC (II)** | $11458.97 \pm 418.0$ | **$2730.55 \pm 706.18$** | $4090.49 \pm 515.31$ | **$5512.54 \pm 449.47$** |
| SW | $11051.35 \pm 835.13$ | $2665.98 \pm 836.71$ | $3790.18 \pm 1256.97$ | $1796.96 \pm 655.34$ |
| **Our SW (I)** | **$12078.74 \pm 485.44$** | **$3137.91 \pm 581.12$** | **$3928.44 \pm 751.13$** | $1494.72 \pm 564.73$ |
| **Our SW (II)** | **$12091.58 \pm 441.79$** | $2535.24 \pm 762.39$ | **$3866.72 \pm 762.83$** | **$1908.97 \pm 541.95$** |
| MC | $10216.53 \pm 887.95$ | $2842.6 \pm 848.36$ | $4067.06 \pm 368.5$ | $4008.76 \pm 1084.86$ |
| **Our MC (I)** | **$12146.75 \pm 386.0$** | **$3307.54 \pm 531.67$** | **$4748.73 \pm 488.4$** | $1981.49 \pm 345.45$ |
| **Our MC (II)** | **$11314.11 \pm 646.23$** | **$3408.04 \pm 295.1$** | **$4906.55 \pm 497.48$** | $2578.92 \pm 403.54$ |
| N-Step | $9997.67 \pm 507.3$ | $3303.43 \pm 387.69$ | $4043.92 \pm 620.2$ | $4837.39 \pm 1111.23$ |
| **Our N-Step (I)** | **$11296.47 \pm 655.26$** | $2748.11 \pm 915.87$ | **$4958.77 \pm 1022.57$** | **$5481.1 \pm 336.03$** |
| **Our N-Step (II)** | **$10962.0 \pm 598.56$** | **$3467.69 \pm 295.89$** | $4466.6 \pm 314.68$ | **$5712.73 \pm 124.91$** |

represents the average reward accumulated over a predefined number of episodes, commonly depicted as bold curves, where the bold lines denote the mean values across 5 unique seeds, and the shaded regions correspond to the standard deviation. In contrast, **the final return** measures the average reward obtained during the last 100,000 steps of the 1 million-step training process. The aim is to maximize both **the episode return** and **the final return** through the training process.

## 5.2 COMPARISON WITH SOTA METHODS

In this section, we validate our approach by enhancing four double-actor DRL methods and five other SOTA DRL methods, with an emphasis on final returns, as presented in Table 1. This table presents the average reward achieved during the final 100,000 training steps across five random seed runs, labeled as "Mean," along with the corresponding standard deviation of these rewards, indicated as "Standard Deviation."

**(1) Experimental results**. Based on the experimental results, we draw two key conclusions. **First and foremost**, both the first and second implementations of our approach significantly improved the performance of existing DRL methods in most cases, in terms of return. This highlights the effectiveness and generality of both implementations, demonstrating the ability of our method to be seamlessly integrated into and enhance the performance of existing DRL methods. **Secondly**, the second implementation of our method (which leverages the advantage function) outperforms the first implementation in a larger number of cases.

**(2) Analysis of results**. The analysis of the experimental results is as follows: **(2.1) Analysis of the First Conclusion:** Our method promotes mutual learning among actor-critic pairs by enabling them to mimic each other's policies. This approach prevents isolated exploration and allows each actor-critic to benefit from the superior policies of others, thereby enhancing existing double-actor

Table 2: Ablation verification (Mean $\pm$ Standard deviation).

| Methods | HalfCheetah | Hopper | Walker2d | Ant |
|---|---|---|---|---|
| **Our CAL** | **11689.88 $\pm$ 832.88** | **3529.85 $\pm$ 260.04** | **4821.53 $\pm$ 502.77** | **4221.99 $\pm$ 367.92** |
| Our Actor (CAL) | 10803.54 $\pm$ 2040.74 | 3099.21 $\pm$ 790.79 | 4308.35 $\pm$ 358.03 | 2772.72 $\pm$ 744.53 |
| Our Critic (CAL) | 11385.36 $\pm$ 399.34 | 3171.47 $\pm$ 585.42 | 4419.47 $\pm$ 516.02 | 3278.33 $\pm$ 1120.44 |
| **Our MC** | **12146.75 $\pm$ 386.0** | **3307.54 $\pm$ 531.67** | **4748.73 $\pm$ 488.4** | 1981.49 $\pm$ 345.45 |
| Our Actor (MC) | 11499.04 $\pm$ 952.45 | 3190.57 $\pm$ 432.46 | 4101.6 $\pm$ 743.19 | **2107.74 $\pm$ 527.67** |
| Our Critic (MC) | 11555.08 $\pm$ 681.24 | 2788.35 $\pm$ 720.81 | 4600.22 $\pm$ 440.05 | **2286.06 $\pm$ 1121.8** |

DRL methods and achieving higher returns. By integrating our method into other DRL approaches through the incorporation of multiple actor-critic pairs that can both collaborate and compete, we enhance exploration capabilities. As a result, our method also improves the policy performance of other types of DRL, leading to higher returns. **(2.2) Analysis of the Second Conclusion:** In contrast to the first implementation, the second implementation assesses the advantages of each actor's actions to identify the most valuable actions to imitate, thereby avoiding indiscriminate policy imitation. This selective approach frequently results in superior performance. When employing our method, users can evaluate both implementations to identify the most effective approach for their specific scenarios.

### 5.3 ABLATION VERIFICATION

In this section, we perform an ablation analysis of our method. Since our approach does not introduce any new hyperparameters, we did not conduct a hyperparameter sensitivity test. Specifically, as outlined in Section 4.2, our method involves mutual imitation between each pair of actor and critic. When computing the collaborative loss, we minimize the difference between $\phi_k(s_t)$ and $\phi_i(s_t)$, as well as the difference between $Q_k(s_t, \phi_k(s_t))$ and $Q_i(s_t, \phi_i(s_t))$. To assess the individual impact of minimizing these differences, we set up two baselines. One baseline, termed Our Actor, focuses solely on minimizing the difference between $\phi_k(s_t)$ and $\phi_i(s_t)$. The other baseline, termed Our Critic, focuses solely on minimizing the difference between $Q_k(s_t, \phi_k(s_t))$ and $Q_i(s_t, \phi_i(s_t))$. We perform ablation analyses for both the CAL and MC methods and implement our approach using the first method. Consequently, we evaluate four baseline methods: Our Actor (CAL), Our Critic (CAL), Our Actor (MC), and Our Critic (MC).

**(1) Experimental results**. The experimental results, as summarized in Table 2, indicate that the full version of our method consistently outperforms both baseline approaches. For instance, our CAL method surpasses both Our Actor (CAL) and Our Critic (CAL) in four simulation tasks. These results demonstrate that optimal performance is not attained merely by having critics or actors imitate each other in isolation. Instead, the best results are achieved when both the actors and critics simultaneously engage in mutual imitation. **(2) Analysis of results**. The first baseline focuses solely on enabling imitation among the actors. While this approach aligns the actions produced by the actors, it lacks constraints on the critics, leading to potentially significant discrepancies in the Q-value estimates for the imitated actions. This impairs the accurate assessment of each action's true value and degrades the overall performance of our method. The second baseline enforces imitation among the critics only. Although this minimizes discrepancies in the critics' Q-values, it does not promote imitation among the actors. Consequently, the policies learned by the actors may vary significantly, making it difficult for the corresponding Q-values to effectively estimate the true value of each action. This, in turn, reduces the overall performance of our method.

## 6 CONCLUSION AND LIMITATIONS

This work develops a generic solution to promote mutual imitation among the actors in double-actor DRL methods, resulting in enhanced performance. Specifically, we minimize the disparities in actions produced by the actors and the differences in Q-values generated by the critics to facilitate mutual action imitation while mitigating excessive discrepancies in value estimation for the imitated actions. We present two specific implementations of our method and validate their effectiveness through comprehensive experimentation. Nonetheless, one limitation remains for future exploration: the development of a more advanced method for action value assessment beyond the advantage function to filter and imitate higher-quality actions represents a promising direction for further research.

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

## A   APPENDIX

### A.1   METHODOLOGY

In this section, we first present the complete training process of our method, along with a detailed theoretical analysis. Finally, we provide the pseudocode implementation of our approach.

#### A.1.1   TRAINING PROCESS OF OUR METHOD

Given the unique characteristics of various DRL algorithms, we will use TD3 to illustrate the full training process of our method. When adapting our approach to other DRL algorithms, the update rules for the actor and critic will need to be adjusted accordingly.

**(1) Random exploration or Collecting samples**: To prepare for network updates, the agent initially engages with the environment to collect samples. During this exploration phase, the agent selects actions $a_t$ randomly, which are executed to receive a reward $r_t$ and transition to the next state $s_{t+1}$. This process results in a sample $(s_t, a_t, r_t, s_{t+1}, d_t)$ that is stored in the replay buffer $\mathcal{D}$. The agent repeats this process for $N$ steps to gather sufficient data. Once the exploration phase is complete, the agent shifts to the training phase, where it utilizes the current policy to make action decisions based on the observed state rather than selecting actions at random. This action decision-making process follows our designed action selection strategy (as described in Section 4.3). The resulting samples $(s_t, a_t, r_t, s_{t+1}, d_t)$ continue to be stored in the replay buffer for subsequent training.

**(2) Calculating target Q-value**: To update each actor-critic pair, we begin by computing the target Q-value from the sampled mini-batch $\mathcal{D}_0$. The target Q-value is given by: $\hat{Q}(s_t, a_t) \leftarrow r_t + \gamma \min_{i=1,2} \mathcal{T}Q_{\theta^{i'}}(s_{t+1}, \tilde{a})$, where $\tilde{a} \leftarrow \pi_{\phi'}(s_{t+1}) + \epsilon$ with $\epsilon \sim \mathcal{N}(0, 0.2)$. Here, $\phi'$ represents the target actor, while $\theta^{i'}$ denotes the target critic. The term $\mathcal{T}Q_{\theta^{i'}}(s_{t+1}, \tilde{a})$ represents the next Q-value, which can be calculated in various ways depending on the double-actor DRL method employed. For example, Pan et al. Pan et al. (2020) enhance the accuracy of Q-value estimation through a softmax function applied to the next Q-value. When integrating our method with various double-actor DRL methods, it is essential to use their specific methods for computing the next Q-value. After calculating the target Q-value, we sequentially update each actor-critic pair using the following method.

**(3) Updating critic in each actor-critic pair**: Let $\alpha$ denote the learning rate. Initially, we update the critic network for each actor-critic pair (such as the $i^{th}$ critic $\theta^i$) by employing the target Q-value $\hat{Q}(s_t, a_t)$, as detailed below:

$$\theta^i \leftarrow \theta^i + \alpha \nabla_{\theta^i} \frac{1}{|\mathcal{D}_0|} \sum_{(s_t, a_t) \in \mathcal{D}_0} \left( \hat{Q}(s_t, a_t) - Q_{\theta^i}(s_t, a_t) \right)^2 \tag{6}$$

**(4) Updating actor in each actor-critic pair**: After updating the critic $\theta^i$, we then update the corresponding actor (e.g., the $i$-th actor $\phi_i$).

**(4.1) Computing the collaborative loss $L_i$**: For the $i$-th actor, we first calculate the collaborative loss $L_i$ to encourage imitation between different actors (e.g., $\phi_k(s_t)$ and $\phi_i(s_t)$) while avoiding excessive discrepancies in their corresponding Q-values (e.g., $Q_k(s_t, \phi_k(s_t))$ and $Q_i(s_t, \phi_i(s_t))$). We develop two different methods to compute $L_i$.

**First Approach**: The first approach directly uses the difference between $\phi_k(s_t)$ and $\phi_i(s_t)$, and the difference between $Q_k(s_t, \phi_k(s_t))$ and $Q_i(s_t, \phi_i(s_t))$ as the collaborative loss $L_i$, aiming to minimize the discrepancy between them, as follows:

$$L_i = \sum_{\substack{k=1 \ k \neq i}}^{n} \left( \frac{1}{|\mathcal{D}_0|} \sum_{i=1}^{|\mathcal{D}_0|} \mathrm{Fs}(Q_k\left(s_t, \phi_k(s_t)\right) - Q_i\left(s_t, \phi_i(s_t)\right) \cdot \left(\phi_k(s_t) - \phi_i(s_t)\right)^2 \right) \quad (7)$$

where $\mathrm{Fs}(x) = \frac{1}{2}\left(\mathrm{sgn}(x) + 1\right)$ is used to map the difference between Q-values to a range of $[0, 1]$ to prevent it from becoming too large.

**Second Approach**: The second approach does not compute the difference between the actions $\phi_k(s_t)$ and $\phi_i(s_t)$ directly as the loss $L_i$. Instead, it estimates the advantage functions for each action to determine whether $\phi_i(s_t)$ should mimic $\phi_k(s_t)$, as the advantage function evaluates how actions $\phi_k(s_t)$ and $\phi_i(s_t)$ perform relative to the average, with a higher advantage indicating a potentially higher value. Specifically, if the advantage of action $\phi_k(s_t)$ is greater than the advantage of action $\phi_i(s_t)$ of the actor $i$ being updated, then $\phi_i(s_t)$ should mimic $\phi_k(s_t)$, thus minimizing the difference between them. Conversely, if the advantage of $\phi_k(s_t)$ is less than that of $\phi_i(s_t)$, there is no need to mimic the action $\phi_k(s_t)$.

Let $\theta^v$ be the network for outputting the value function $V(s)$. The advantage functions for two actions are given by $A(s, \phi_i(s_t))$ and $A(s, \phi_k(s_t))$, where: $A(s, \phi_i(s_t)) = Q_i(s_t, \phi_i(s_t)) - V(s)$ and $A(s, \phi_k(s_t)) = Q_i(s_t, \phi_k(s_t)) - V(s)$. Next, we compare the magnitudes of these advantage functions as follows:

$$\hat{a} = \phi_i(s_t) * \mathbf{1}[A(s, \phi_i(s_t)) >= A(s, \phi_k(s_t))] + \phi_k(s_t) * \mathbf{1}[A(s, \phi_k(s_t)) >= A(s, \phi_i(s_t))] \quad (8)$$

where $\hat{a}$ represents the action that $\phi_i(s_t)$ needs to mimic. The function $\mathbf{1}[x, y]$ is an indicator function that equals 1 if $x > y$ and 0 otherwise. Subsequently, similar to the first method, we calculate the collaborative loss $L_i$, given by:

$$L_i = \sum_{\substack{k=1 \ k \neq i}}^{n} \left( \frac{1}{|\mathcal{D}_0|} \sum_{i=1}^{|\mathcal{D}_0|} \mathrm{Fs}(Q_k\left(s_t, \phi_k(s_t)\right) - Q_i\left(s_t, \phi_i(s_t)\right) \cdot \left(\hat{a} - \phi_i(s_t)\right)^2 \right) \quad (9)$$

For updating the value function network $\theta^v$, we adopt the method used in existing literature. We start by calculating the next Q-value $\tilde{Q}(s_t, a_t)$ with the current critics: $\tilde{Q}(s_t, a_t) \leftarrow r_t + \gamma \min_{i=1,2} \mathcal{T}Q_{\theta^i}(s_{t+1}, \tilde{a})$ where $\tilde{a} \leftarrow \pi_{\phi'}(s_{t+1}) + \epsilon$, with $\epsilon \sim \mathcal{N}(0, 0.2)$. Here, $\theta^i$ denotes the current critic. The target value $\hat{V}(s)$ is then computed as: $\hat{V}(s) = \min\left(\tilde{Q}(s_t, a_t), \hat{Q}(s_t, a_t)\right)$. The value function network $\theta^v$ is updated by minimizing the difference between the target value and the current value, as follows:

$$\mathcal{L}_V(\theta^v) = \mathbb{E}_{(s,a) \sim \mathcal{D}_0}\left[\left(\hat{V}(s) - V(s)\right)^2\right] \quad (10)$$

**(4.2) Computing the gradient of the actor**: After calculating the collaborative loss $L_i$, we update the $i^{th}$ actor $\phi_i$ using $Q_{\theta^i}(s_t, \pi_{\phi_i}(s_t))$, $L_i$, and $\pi_{\phi_i}(s_t)$ as follows:

$$\nabla_{\phi_i} J(\phi_i) = \frac{1}{|\mathcal{D}_0|} \sum_{(s_t, a_t) \in \mathcal{D}_0} \nabla_{\pi_{\phi_i}(s_t)} Q_{\theta^i}(s_t, \pi_{\phi_i}(s_t)) \nabla_{\phi_i} \pi_{\phi_i}(s_t) + \nabla_{\pi_{\phi_i}(s_t)} L_i \quad (11)$$

**(5) Updating the target networks**: Once the actor-critic pair is updated, we proceed to update the corresponding target actor-critic pair using a soft update method, which entails applying updates with a two-step delay. Specifically, these updates are defined as follows:

$$\phi_i' \leftarrow \tau\phi_i + (1-\tau)\phi_i'; \theta^{i'} \leftarrow \tau\theta^i + (1-\tau)\theta^{i'} \tag{12}$$

where $\tau = 0.005$ denotes the coefficient for the soft update. These updates are applied repeatedly throughout the training process. Furthermore, our method integrates seamlessly with existing double-actor DRL methods, preserving their convergence properties and Q-value accuracy, as it does not impact the Q-value update.

### A.1.2 THEORETICAL ANALYSIS

This section provides a theoretical analysis demonstrating why our method advances beyond existing double-actor DRL methods. We start by introducing an assumption and five lemmas that form the basis for deriving Theorem 1. Finally, we discuss why our approach enhances current double-actor DRL methods.

**Assumption 1**: In DRL, the reward signal $r(s)$ is constrained such that its absolute value does not exceed a constant $R_{\max}$. This can be expressed as $|r(s)| \leq R_{\max}$.

**Lemma 1**: As outlined in the study Xiong et al. (2022), the objective function $J(\pi(\phi))$ in DRL is given by:

$$J(\pi(\phi)) = \frac{1}{1-\gamma}\mathbb{E}_{s \sim D(\pi(\phi))}[r(s)] \tag{13}$$

In this expression, $\frac{1}{1-\gamma}$ is a positive constant applicable to all policies, with $\gamma$ representing the discount factor used in DRL. $D(\pi(s))$ represents the occupancy measure for policy $\pi$, as defined by Ran et al. (2023). The occupancy measure $D(\pi(s'))$ is given by:

$$D(\pi(s')) = \int_{\mathcal{S}} \sum_{t=0}^{\infty} (1-\gamma)\gamma^t p_0(s) \, p(s \rightarrow s', t, \pi) \, \mathrm{d}s \tag{14}$$

Here, $p(s \rightarrow s', t, \pi)$ denotes the probability density of moving from state $s$ to state $s'$ after $t$ time steps under policy $\pi$.

**Lemma 2**: In the study Ran et al. (2023), the expectation $\mathbb{E}_{s \sim D(\pi(s))}[r(s)]$ is formulated as follows:

$$\mathbb{E}_{s \sim D(\pi(s))}[r(s)] = \int_{\mathcal{S}} r(s) \, D(\pi(s))(s) \, \mathrm{d}s \tag{15}$$

**Lemma 3**: In Xiong et al. (2022), it is demonstrated that for any two actions $\pi_1(s)$ and $\pi_2(s)$ within the action space $\mathcal{A}$, a positive constant $K_c$ exists such that:

$$\int_{\mathcal{S}} |D(\pi_1(s)) - D(\pi_2(s))| \, \mathrm{d}s \leq K_c \max_{s \in \mathcal{S}} \|\pi_1(s) - \pi_2(s)\| \tag{16}$$

**Lemma 4**: Consider a function $f : S \subset \mathbb{R}^m \rightarrow \mathbb{R}$. If $p \leq q$ and the interval $[p, q]$ is contained within $S$, then the following inequality is valid:

$$\left| \int_p^q f(x)\mathrm{d}x \right| \leq \int_p^q |f(x)|\mathrm{d}x \tag{17}$$

Proof. We can consider two cases: when the integral is nonnegative and when it is negative.

Case 1: Assume that $\int_a^b f(x)\mathrm{d}x \geq 0$. In this case, we have

$$\left| \int_a^b f(x) \, dx \right| = \int_a^b f(x) \, dx \quad \text{(since the integral is nonnegative)}$$

$$\leq \int_a^b |f(x)| \, dx \quad \text{(since } |f(x)| \geq f(x) \text{ for all } x)$$

Case 2: Assume that $\int_a^b f(x) dx < 0$. In this case, we have

$$\left| \int_a^b f(x) \, dx \right| = -\int_a^b f(x) \, dx \quad \text{(since the integral is negative)}$$

$$= \int_a^b -f(x) \, dx$$

$$\leq \int_a^b |f(x)| \, dx \quad \text{(since } -f(x) \leq |f(x)| \text{ for all } x)$$

Therefore, in both cases, we have shown that

$$\left| \int_a^b f(x) dx \right| \leq \int_a^b |f(x)| \tag{18}$$

which completes the proof of the inequality.

**Lemma 5**: To bound the difference between the long-term cumulative rewards $J(\pi_1(s))$ and $J(\pi_2(s))$ for policies $\pi_1(s)$ and $\pi_2(s)$, consider the following:

$$
\begin{aligned}
&|J(\pi_1(s)) - J(\pi_2(s))| \\
&\stackrel{(Lemma1)}{=} \frac{1}{1-\gamma} \left| \mathbb{E}_{s \sim D(\pi_1(s))}[r(s)] - \mathbb{E}_{s \sim D(\pi_2(s))}[r(s)] \right| \\
&\stackrel{(Lemma2)}{=} \frac{1}{1-\gamma} \left| \int_{\mathcal{S}} r(s) \left( D(\pi_1(s)) - D(\pi_2(s)) \right) \, ds \right| \\
&\stackrel{(Lemma4)}{\leq} \frac{1}{1-\gamma} \int_{\mathcal{S}} |r(s) \left( D(\pi_1(s)) - D(\pi_2(s)) \right)| \, ds \\
&= \frac{1}{1-\gamma} \int_{\mathcal{S}} |r(s)| \left| \left( D(\pi_1(s)) - D(\pi_2(s)) \right) \right| \, ds \\
&\stackrel{(Assumption1)}{\leq} \frac{R_{\max}}{1-\gamma} \int_{\mathcal{S}} |D(\pi_1(s)) - D(\pi_2(s))| \, ds \\
&\stackrel{(Lemma3)}{\leq} \frac{R_{\max} K_c}{1-\gamma} \max_{s \in \mathcal{S}} \|\pi_1(s) - \pi_2(s)\|
\end{aligned}
\tag{19}
$$

This expression indicates that the disparity between the two policies directly affects the difference in cumulative reward $J$, with $\frac{R_{\max} K_c}{1-\gamma}$ serving as a positive constant across all policies.

**Theorem 1**: Let $n$ represent the number of actor-critic pairs in our method, and let $\pi^k(s)$ denote the policy of the $k$-th actor. The difference between $J(\pi^i(s))$, which indicates the performance of the $i$-th actor's policy $\pi^i(s)$, and the optimal policy associated with this policy, $J(\pi_i^*(s))$, can be bounded as follows:

$$|J(\pi_i^*(s)) - J(\pi^i(s))|$$
$$= |J(\pi_i^*(s)) + J(\pi^n(s)) - J(\pi^n(s)) - J(\pi^i(s))|$$
$$\leq |J(\pi_i^*(s)) - J(\pi^n(s))| + |J(\pi^n(s)) - J(\pi^i(s))|$$
$$= |J(\pi_i^*(s)) + J(\pi^{n-1}(s)) - J(\pi^{n-1}(s)) - J(\pi^n(s))| + |J(\pi^n(s)) - J(\pi^i(s))|$$
$$\leq |J(\pi_i^*(s)) - J(\pi^{n-1}(s))| + |J(\pi^{n-1}(s)) - J(\pi^n(s))| + |J(\pi^n(s)) - J(\pi^i(s))|$$
$$...$$
$$= |J(\pi_i^*(s)) - J(\pi^1(s))| + \sum_{k=2 \ k\neq i}^{n} \left( |J(\pi^{k-1}(s)) - J(\pi^k(s))| \right) + |J(\pi^n(s)) - J(\pi^i(s))| \tag{20}$$
$$\overset{(Lemma5)}{\leq} \frac{R_{\max}K_c}{1-\gamma} \left( \max_{s\in\mathcal{S}} \|\pi_i^*(s) - \pi^1(s)\| + \sum_{k=2 \ k\neq i}^{n} \left( \max_{s\in\mathcal{S}} \|\pi^{k-1}(s) - \pi^k(s)\| \right) \right)$$
$$+ \max_{s\in\mathcal{S}} \|\pi^n(s) - \pi^i(s)\| \Big)$$

**Discussion**: The above derivation demonstrates that facilitating mutual imitation among the policies of the actor-critic pairs can reduce $\sum_{k=2 \ k\neq i}^{n} \left( \max_{s\in\mathcal{S}} \|\pi^{k-1}(s) - \pi^k(s)\| \right) + \max_{s\in\mathcal{S}} \|\pi^n(s) - \pi^i(s)\|$, thereby lowering the upper bound on the difference between the policy $J(\pi^i(s))$ and the optimal policy $J(\pi_i^*(s))$. Consequently, the performance $J$ of $\pi^i(s)$ has the potential to more closely align with that of the optimal policy $\pi_i^*(s)$. We apply the same analysis, as outlined in Theorem 1, to each actor in the remaining actor-critic pairs within our method. This analysis suggests that each actor's policy will potentially converge more closely to its optimal policy. Therefore, by facilitating mutual imitation between actors and critics, our approach enhances the policy performance of existing double-actor DRL methods.

### A.1.3 PSEUDOCODE IMPLEMENTATION

In this section, we provide a pseudocode implementation of our approach, using the TD3 algorithm as an illustrative example, as detailed in Algorithm 1 (Implementation I) and Algorithm 2 (Implementation II). This pseudocode details the training procedure specific to TD3. However, since training methods can vary across different DRL techniques, it is essential to adapt the pseudocode by integrating the specific components of each DRL method as required.

### A.2 EXPERIMENT AND ANALYSIS

This section details the implementation specifics and the episode returns from our experiments. The results are depicted by the bold curve, representing the average performance, while the shaded region illustrates the standard deviation across five runs.

### A.2.1 IMPLEMENTATION DETAILS

In this section, we provide detailed implementation specifics for both the underlying DRL algorithm and our method. Implementation details for baseline DRL methods are available in the existing literature and are not reiterated here.

**Implementation of the Underlying DRL Algorithm**. We use a multilayer perceptron architecture consisting of three layers: two hidden layers and one output layer, which is consistent across all DRL algorithms in this study. Both the actor and critic modules feature hidden layers with 256 neurons each, utilizing the ReLU activation function. In contrast, the actor's output layer employs the Tanh activation function. A fixed learning rate of 0.001 is used for both the actor and critic, with the Adam optimizer managing the training process. We train the model using mini-batches of 256 samples and maintain a replay buffer with a capacity of 1,000,000. The discount factor is set to 0.99, and the target network undergoes a soft update every 2 steps at a rate of 0.005. The training process extends over 1 million steps, with the initial 10,000 steps reserved for random exploration. Thus, as outlined in Algorithm 1 and Algorithm 2, the number of effective training steps is 990,000.

---

**Algorithm 1** Training procedure for our method (Implementation I)

---

1: Initialize $n$ actor-critic pairs $(\phi_i, \theta^i)$, along with their target networks $(\phi^{i'}, \theta^{i'})$.
2: Initialize the replay buffer $\mathcal{D}$
3: Initialize a random process for action exploration
4: Initialize the maximum number of training steps $M$
5: $t = 1, p = 1$
6: **repeat**
7:     Utilize each actor $i$ to generate each action as $a_t^i = \text{clip}(\mu(s_t|\phi_i) + \epsilon)$.
8:     Utilize each critic $\theta^i$ to evaluate the Q-value $Q_i\left(s_t, a_t^i\right)$ for the corresponding action $a_t^i$.
9:     Select the action $a_t$ with the larger Q-value
10:    Execute the selected action $a_t$ and observe the resulting reward $r_t$ and the new state $s_{t+1}$
11:    Store transition $(s_t, a_t, r_t, s_{t+1})$ in the replay buffer $\mathcal{D}$
12:    Sample the mini-batch $\mathcal{D}_0$ from the replay buffer $\mathcal{D}$
13:    Compute the target Q-value with $\mathcal{D}_0$: $\hat{Q}_p\left(s_i, a_i\right) = r_i + \gamma \min_{j=1,2} Q'_p(s_{i+1}, \mu'(s_{i+1}|\phi'_p)|\theta^{Q'_p})$
14:    **repeat**
15:       Update critic $\theta^{Q_p}$ by minimizing the loss: $\mathcal{L} = \frac{1}{|\mathcal{D}_0|} \sum_{i=1}^{|\mathcal{D}_0|} (\hat{Q}_p\left(s_i, a_i\right) - Q_p(s_i, a_i|\theta^{Q_p}))^2$
16:       Compute the cooperative loss $L_p$.
17:       Update actor $\phi_p$ with the gradient: $J = |\mathcal{D}_0|^{-1} \sum_{(s,a)\in\mathcal{D}_0} \nabla_a Q\theta^{Q_p}(s, \pi_{\phi(s)_p})\nabla_{\phi_p}\pi_{\phi(s)_p} + \nabla_{\phi_p} L_p$
18:       Every $d$ steps, update target networks: $\theta^{Q'_p} \leftarrow \tau\theta^{Q_p} + (1-\tau)\theta^{Q'_p}, \phi'_p \leftarrow \tau\phi_p + (1-\tau)\phi'_p$
19:       $p++$
20:    **until** $p > n$
21:    $t++$
22: **until** $M - t = 0$

---

**Implementation of Our Method**. Both implementations of our framework do not introduce new hyperparameters. However, the second implementation incorporates a value network to predict the value function. This network processes the state through five fully connected layers: the first four layers each consist of 256 neurons and use the ReLU activation function, while the final layer maps the output to a single dimension representing the predicted value function. The value network is trained using the Adam optimizer with a learning rate of 0.001.

### A.2.2 COMPARISON WITH SOTA METHODS

Our evaluation starts with a total of 1,000,000 training steps ($T_{\max}$) per test unit. The first 10,000 steps of each test unit are dedicated to random exploration. After this initial phase, we conduct 10 evaluation episodes at 10,000-step intervals. Therefore, each curve consists of a total of 100 data points.

**Experimental results**. The experimental results, in terms of episode return, are presented in Fig. 2 through Fig. 10. Based on the experimental results, we draw two key conclusions. **First and foremost**, both the first and second implementations of our approach significantly improved the performance of existing DRL methods in most cases, in terms of return. This highlights the effectiveness and generality of both implementations, demonstrating the ability of our method to be seamlessly integrated into and enhance the performance of existing DRL methods. **Secondly**, the second implementation of our method (which leverages the advantage function) outperforms the first implementation in a larger number of cases.

### A.2.3 ABLATION ANALYSIS

In this section, we perform an ablation analysis of our method. Since our approach does not introduce any new hyperparameters, we did not conduct a hyperparameter sensitivity test. Specifically, as outlined in Section 4.2, our method involves mutual imitation between each pair of actor and critic. When computing the collaborative loss, we minimize the difference between $\phi_k(s_t)$ and $\phi_i(s_t)$, as well as the difference between $Q_k(s_t, \phi_k(s_t))$ and $Q_i(s_t, \phi_i(s_t))$. To assess the individual impact

---

**Algorithm 2** Training procedure for our method (Implementation II)

---

1: Initialize $n$ actor-critic pairs $(\phi_i, \theta^i)$, along with their target networks $(\phi^{i'}, \theta^{i'})$.
2: Initialize the value network $\theta^v$
3: Initialize the replay buffer $\mathcal{D}$
4: Initialize a random process for action exploration
5: Initialize the maximum number of training steps $M$
6: $t = 1, p = 1$
7: **repeat**
8:     Utilize each actor $i$ to generate each action as $a_t^i = \text{clip}(\mu(s_t|\phi_i) + \epsilon)$.
9:     Utilize each critic $\theta^i$ to evaluate the Q-value $Q_i(s_t, a_t^i)$ for the corresponding action $a_t^i$.
10:    Select the action $a_t$ with the larger Q-value
11:    Execute the selected action $a_t$ and observe the resulting reward $r_t$ and the new state $s_{t+1}$
12:    Store transition $(s_t, a_t, r_t, s_{t+1})$ in the replay buffer $\mathcal{D}$
13:    Sample the mini-batch $\mathcal{D}_0$ from the replay buffer $\mathcal{D}$.
14:    Compute the target Q-value with $\mathcal{D}_0$:     $\hat{Q}_p(s_i, a_i)$     $=$     $r_i$     $+$ $\gamma \min_{j=1,2} Q'_p(s_{i+1}, \mu'(s_{i+1}|\phi'_p)|\theta^{Q'_p})$
15:    Compute the current state value $V(s)$ and the target state value $\hat{V}(s)$
16:    Update the value network $\theta^v$ by minimizing $\mathbb{E}_{(s,a)\sim\mathcal{D}_0}\left[\left(\hat{V}(s) - V(s)\right)^2\right]$.
17:    **repeat**
18:        Update critic $\theta^{Q_p}$ by minimizing the loss: $\mathcal{L} = \frac{1}{|\mathcal{D}_0|}\sum_{i=1}^{|\mathcal{D}_0|}(\hat{Q}_p(s_i, a_i) - Q_p(s_i, a_i|\theta^{Q_p}))^2$
19:        Compute the advantage function $A(s, \phi_p(s_t))$ for each action $a_p$.
20:        Compute the cooperative loss $L_p$.
21:        Update actor $\phi_p$ with the gradient: $J = |\mathcal{D}_0|^{-1}\sum_{(s,a)\in\mathcal{D}_0}\nabla_a Q\theta^{Q_p}(s, \pi_{\phi(s)_p})\nabla_{\phi_p}\pi_{\phi(s)_p} + \nabla_{\phi_p}L_p$
22:        Every $d$ steps, update target networks: $\theta^{Q'_p} \leftarrow \tau\theta^{Q_p} + (1-\tau)\theta^{Q'_p}, \phi'_p \leftarrow \tau\phi_p + (1-\tau)\phi'_p$
23:        $p++$
24:    **until** $p > n$
25:    $t++$
26: **until** $M - t = 0$

---

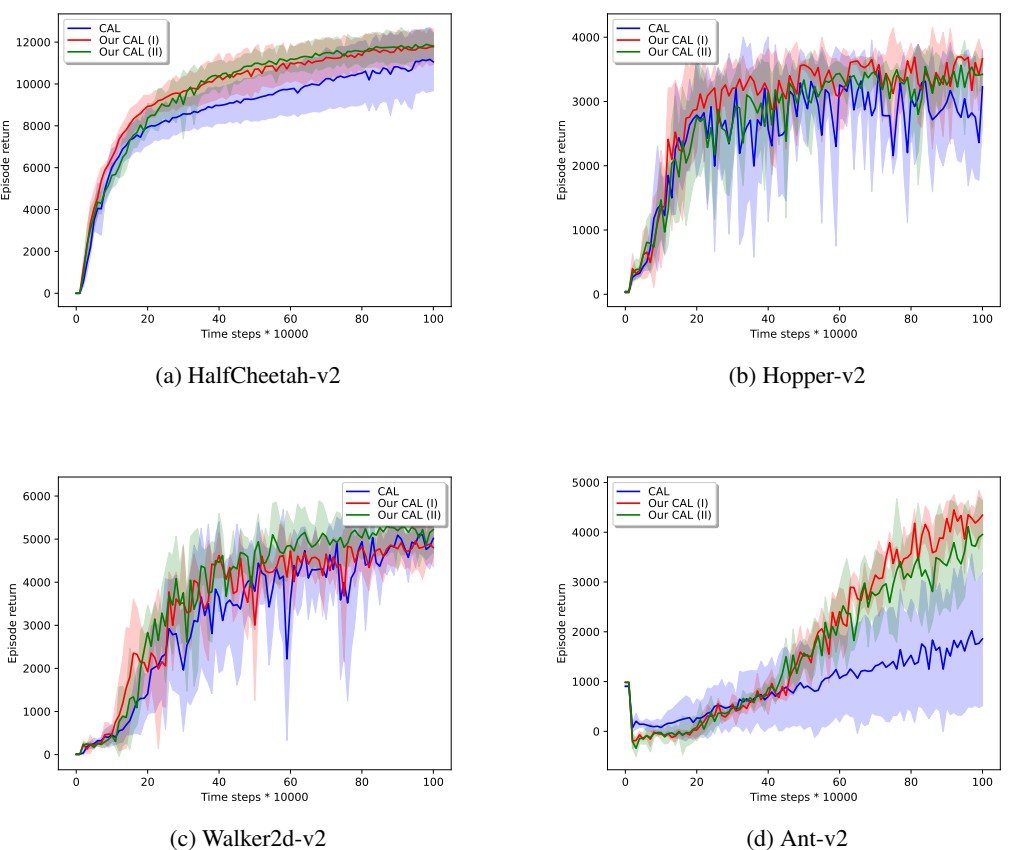

(a) HalfCheetah-v2

(b) Hopper-v2

(c) Walker2d-v2

(d) Ant-v2

Figure 2: Comparison with CAL in terms of episode return across five different runs. The results, shown from top to bottom and left to right, correspond to HalfCheetah-v2, Hopper-v2, Walker2d-v2, and Ant-v2.

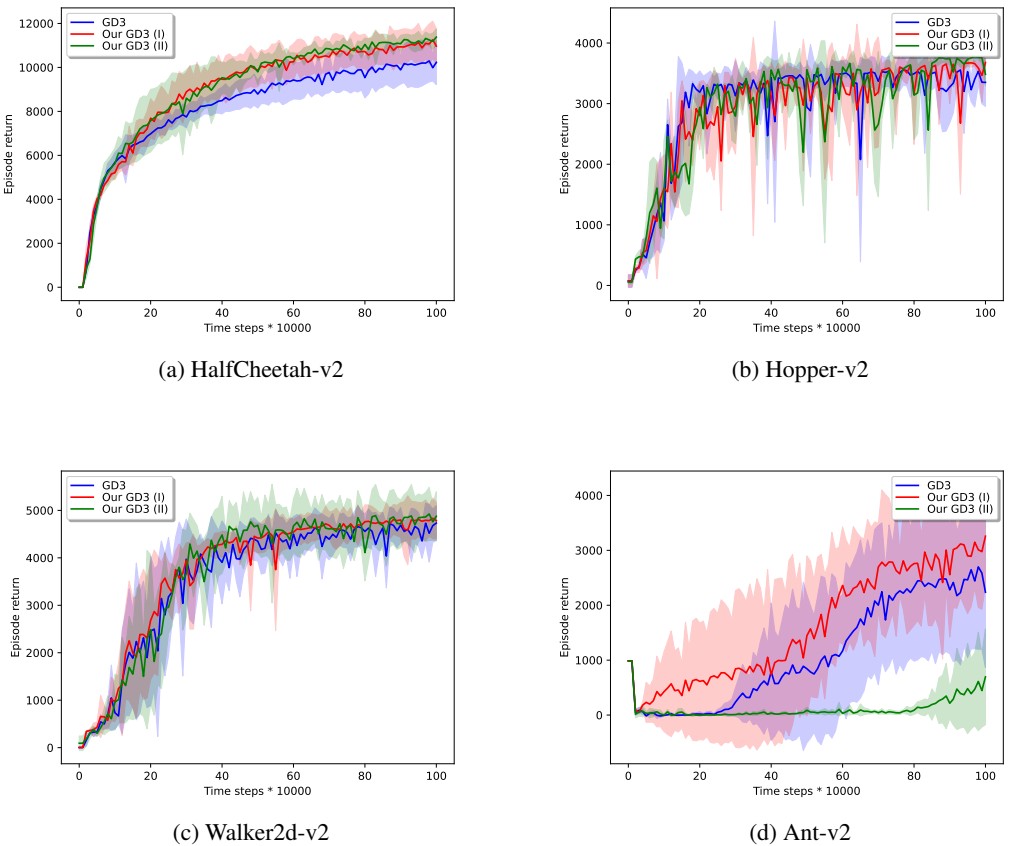

Figure 3: Comparison with GD3 in terms of episode return across five different runs. The results, shown from top to bottom and left to right, correspond to HalfCheetah-v2, Hopper-v2, Walker2d-v2, and Ant-v2.

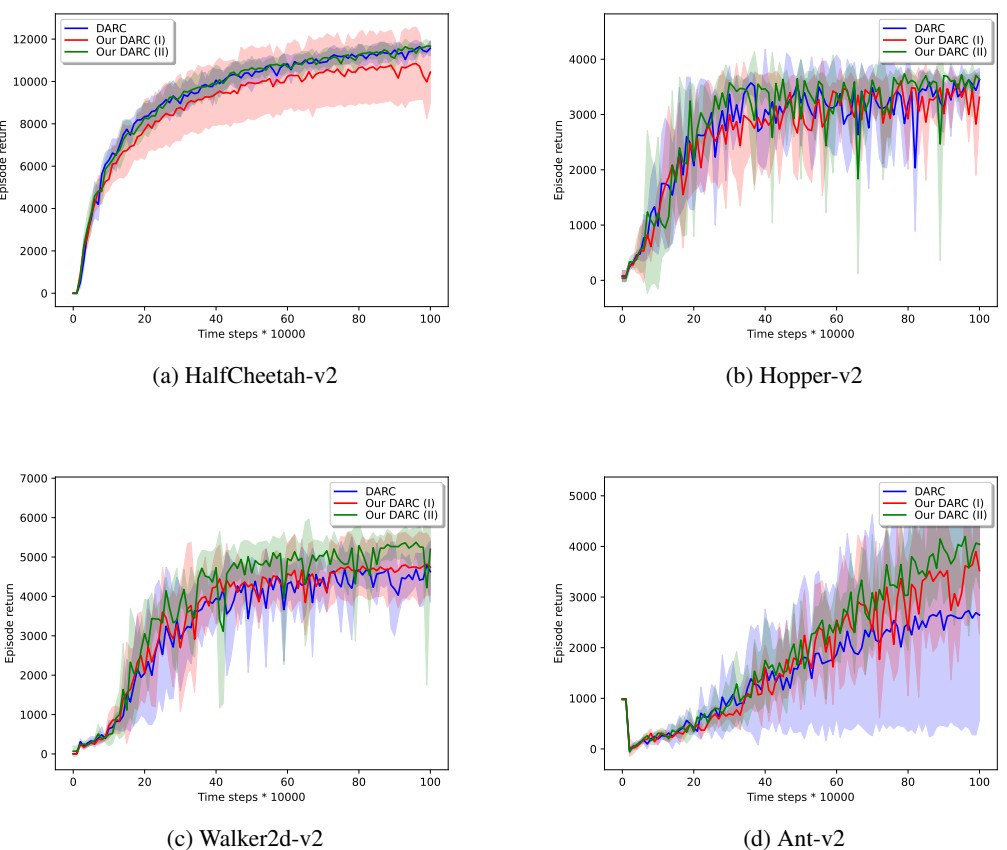

Figure 4: Comparison with DARC in terms of episode return across five different runs. The results, shown from top to bottom and left to right, correspond to HalfCheetah-v2, Hopper-v2, Walker2d-v2, and Ant-v2.

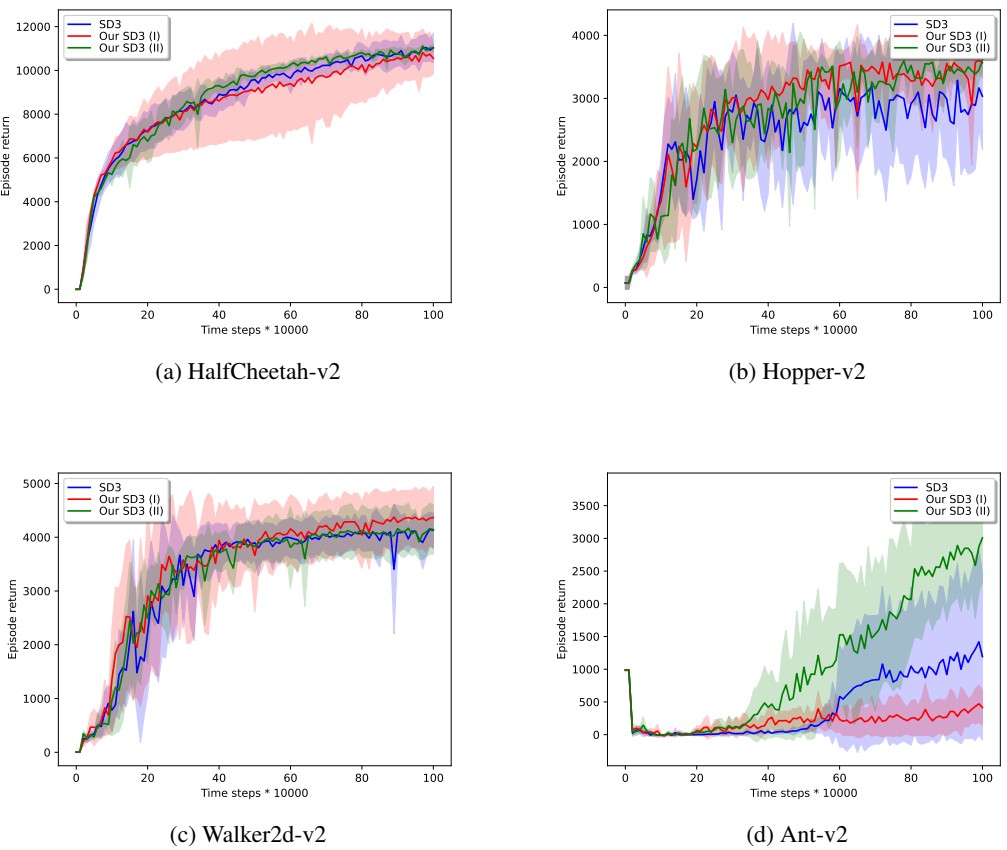

Figure 5: Comparison with SD3 in terms of episode return across five different runs. The results, shown from top to bottom and left to right, correspond to HalfCheetah-v2, Hopper-v2, Walker2d-v2, and Ant-v2.

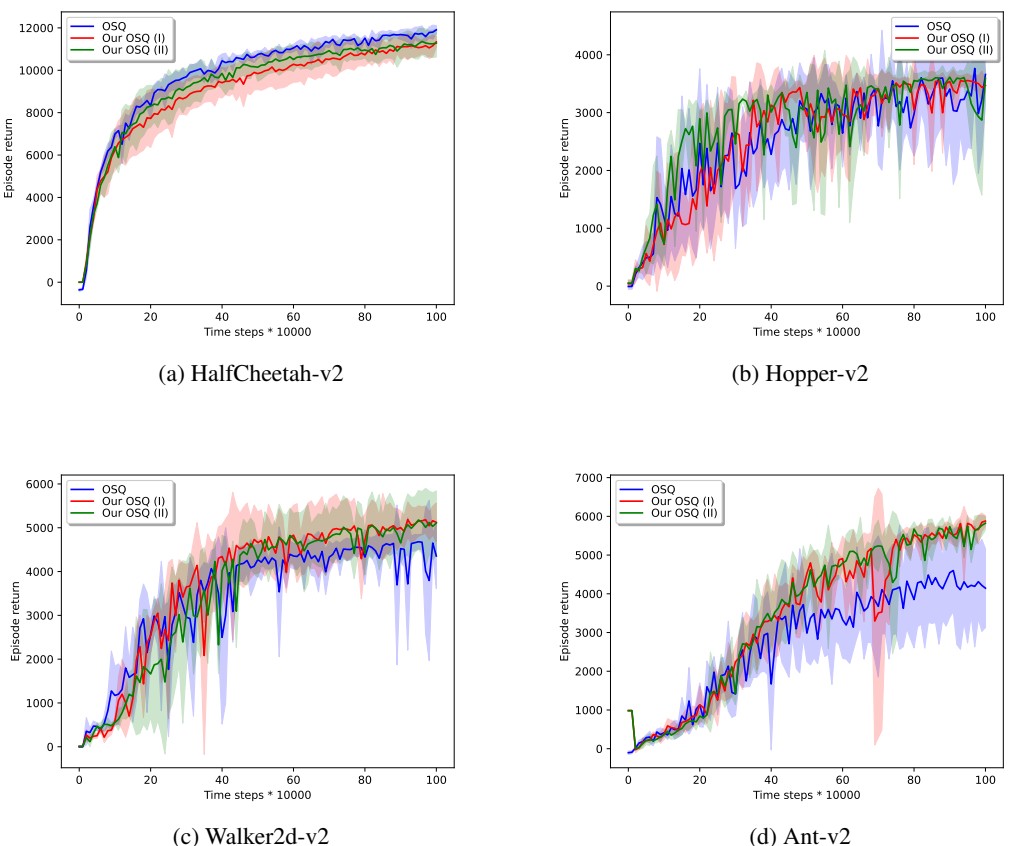

Figure 6: Comparison with OSQ in terms of episode return across five different runs. The results, shown from top to bottom and left to right, correspond to HalfCheetah-v2, Hopper-v2, Walker2d-v2, and Ant-v2.

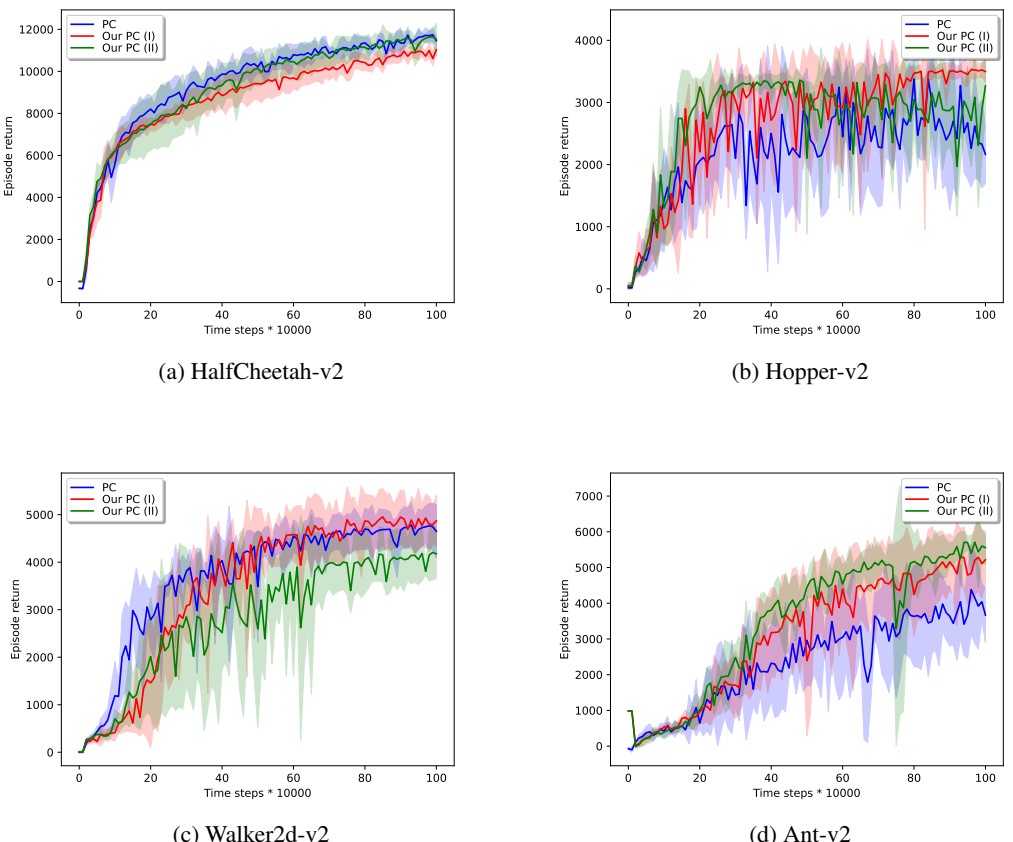

(a) HalfCheetah-v2

(b) Hopper-v2

(c) Walker2d-v2

(d) Ant-v2

Figure 7: Comparison with PC in terms of episode return across five different runs. The results, shown from top to bottom and left to right, correspond to HalfCheetah-v2, Hopper-v2, Walker2d-v2, and Ant-v2.

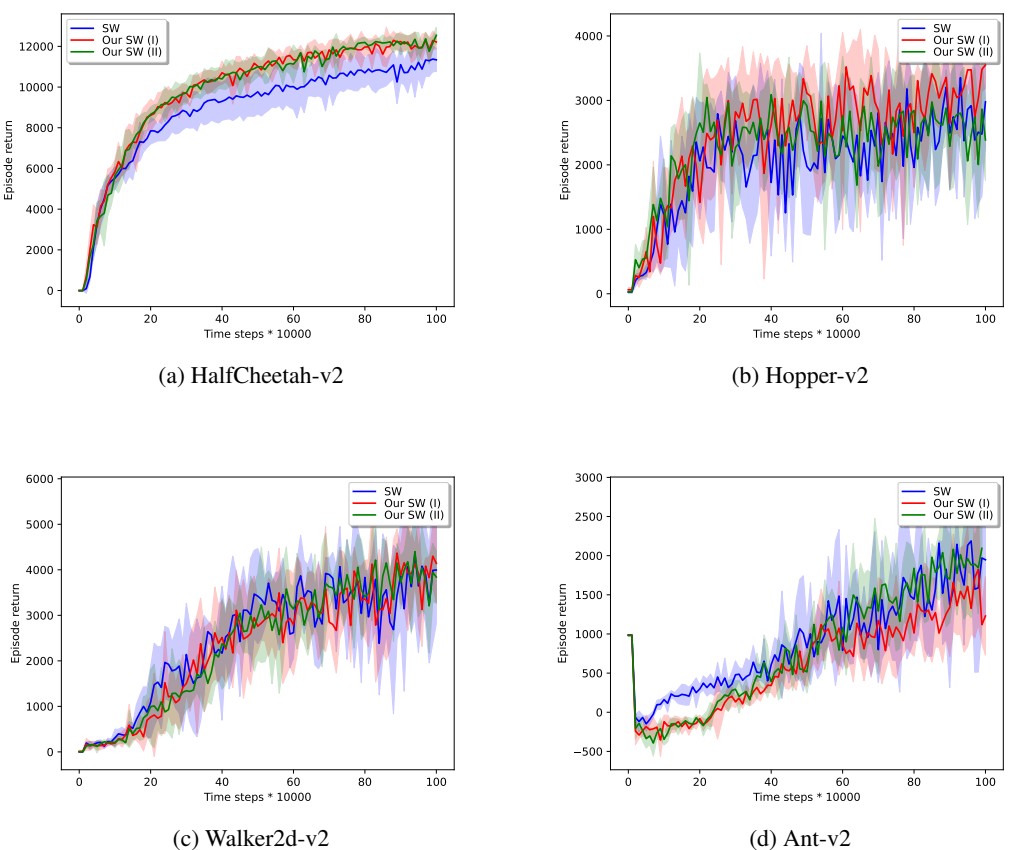

Figure 8: Comparison with SW in terms of episode return across five different runs. The results, shown from top to bottom and left to right, correspond to HalfCheetah-v2, Hopper-v2, Walker2d-v2, and Ant-v2.

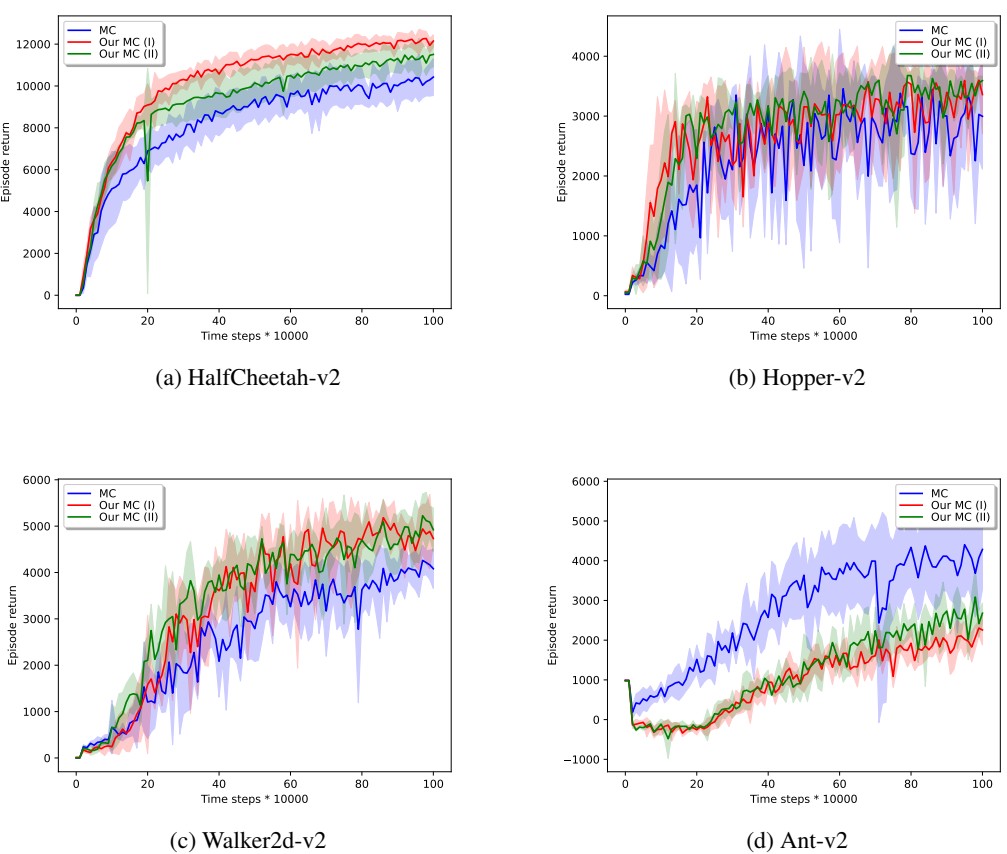

Figure 9: Comparison with MC in terms of episode return across five different runs. The results, shown from top to bottom and left to right, correspond to HalfCheetah-v2, Hopper-v2, Walker2d-v2, and Ant-v2.

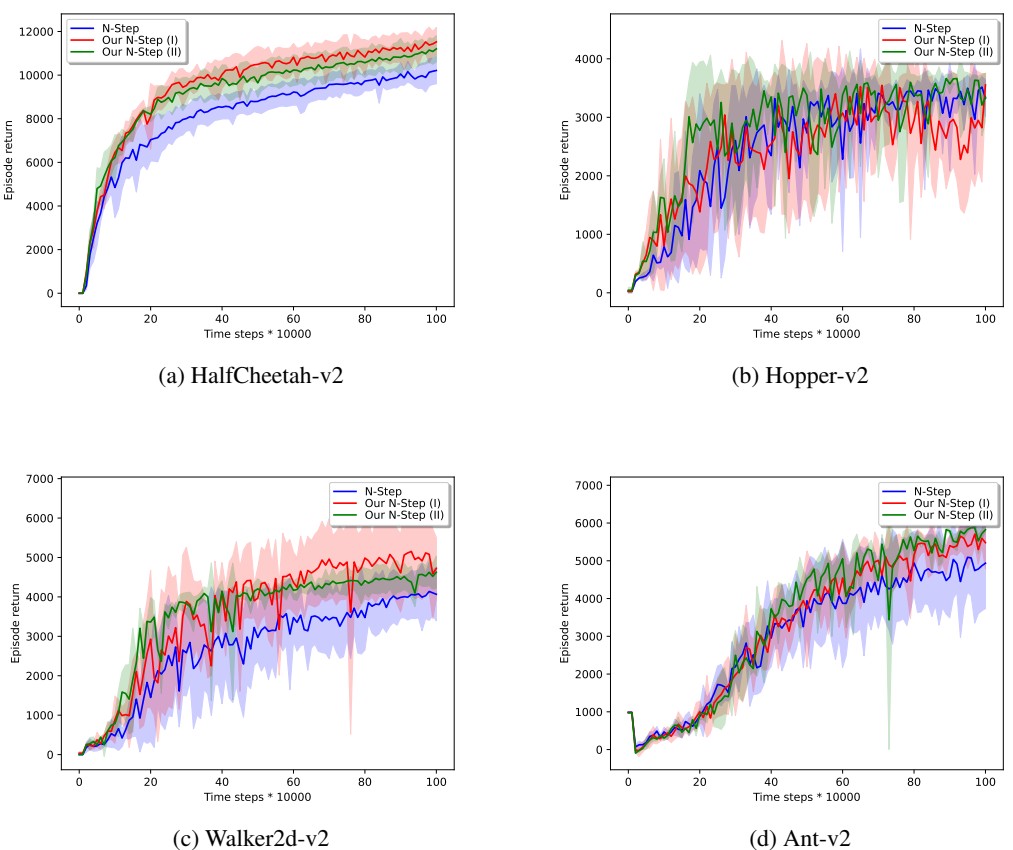

Figure 10: Comparison with N-step in terms of episode return across five different runs. The results, shown from top to bottom and left to right, correspond to HalfCheetah-v2, Hopper-v2, Walker2d-v2, and Ant-v2.

of minimizing these differences, we set up two baselines. One baseline, termed Our Actor, focuses solely on minimizing the difference between $\phi_k(s_t)$ and $\phi_i(s_t)$. The other baseline, termed Our Critic, focuses solely on minimizing the difference between $Q_k(s_t, \phi_k(s_t))$ and $Q_i(s_t, \phi_i(s_t))$. We perform ablation analyses for both the CAL and MC methods and implement our approach using the first method. Consequently, we evaluate four baseline methods: Our Actor (CAL), Our Critic (CAL), Our Actor (MC), and Our Critic (MC).

**Experimental results**. The ablation analysis results, in terms of episode return, are presented in Fig. 11 through Fig. 12. These results indicate that the full version of our method consistently outperforms both baseline approaches. For example, our CAL method exceeds the performance of both Our Actor (CAL) and Our Critic (CAL) across four simulation tasks. This demonstrates that optimal performance is not achieved by having critics or actors imitate each other in isolation. Instead, the best results are obtained when both actors and critics engage in mutual imitation simultaneously.

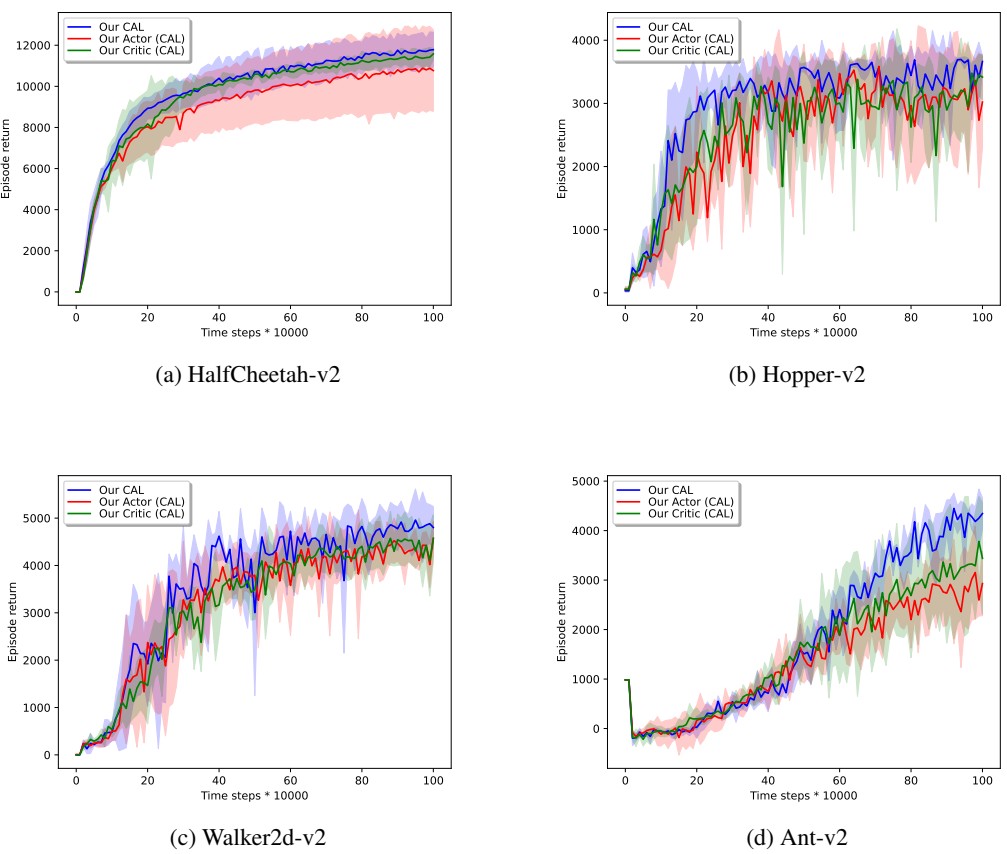

(a) HalfCheetah-v2  (b) Hopper-v2

(c) Walker2d-v2  (d) Ant-v2

Figure 11: Ablation analysis of CAL in terms of episode return across five different runs. The results are displayed from top to bottom and left to right, corresponding to the environments HalfCheetah-v2, Hopper-v2, Walker2d-v2, and Ant-v2.

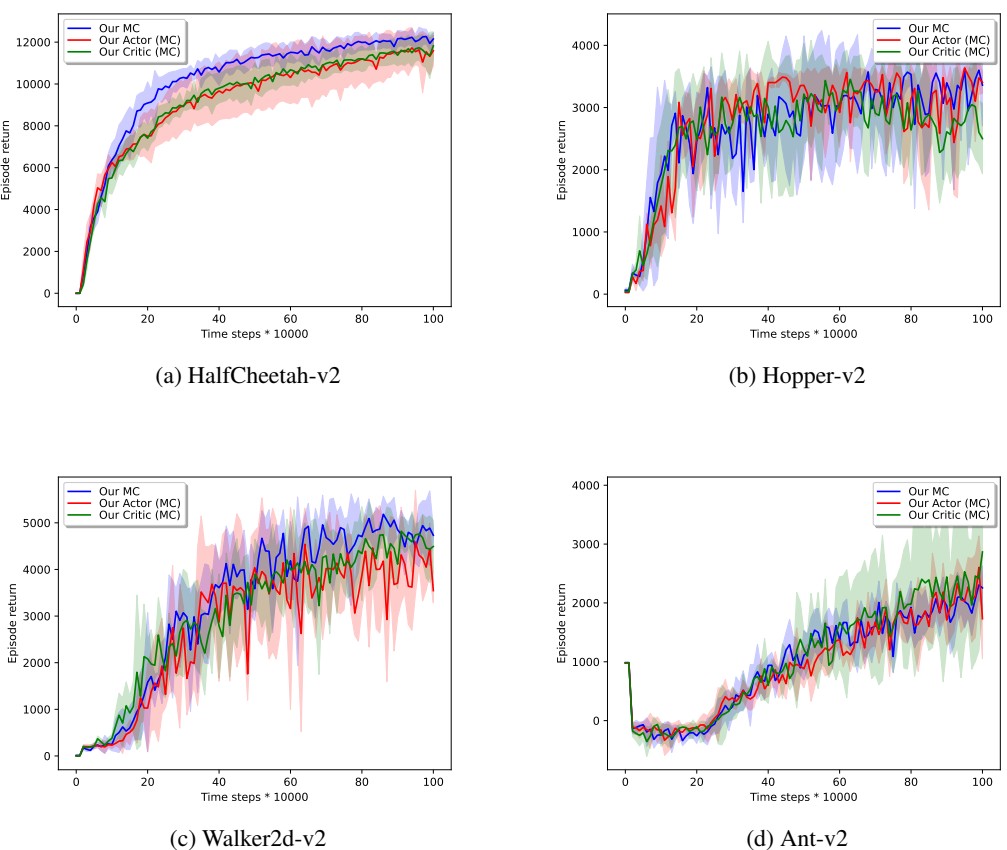

Figure 12: Ablation analysis of MC in terms of episode return across five different runs. The results are displayed from top to bottom and left to right, corresponding to the environments HalfCheetah-v2, Hopper-v2, Walker2d-v2, and Ant-v2.

