# OpenReview forum: "A Competitive-Cooperative Actor-critic Framework for Reinforcement Learning"
_ICLR.cc/2025/Conference — ICLR 2025 Conference Withdrawn Submission_

### Official Review · Reviewer_yMU6 · 2024-10-28

**Soundness:** 3
**Presentation:** 3
**Contribution:** 3
**Rating:** 6
**Confidence:** 5

**Summary:**

This paper proposes a framework that promotes mutual learning and imitation between actors. It achieves this by minimizing differences in actions between actors and Q-value discrepancies between critics, thereby improving the performance of reinforcement learning algorithms. The framework is implemented through two specific approaches: Direct Imitation and Selective Imitation. Direct Imitation minimizes differences in actions produced by actors and Q-value discrepancies between corresponding critics, fostering mutual learning between actors. Selective Imitation utilizes the value function to assess actions and imitates only those with higher assessments, thereby avoiding the replication of lower-quality strategies. Additionally, the framework extends to multi-critic architectures. Experimental results demonstrate that this method significantly improves the performance of nine state-of-the-art reinforcement learning algorithms across four MuJoCo tasks.

**Strengths:**

1. This paper proposes an innovative competitive-cooperative actor-critic framework, which addresses the limitation of independent exploration in existing double-actor methods.
2. This paper rigorously analyzes the proposed framework and validates its effectiveness through extensive experiments.
3. This paper proposes a general and scalable framework for improving the performance of reinforcement learning algorithms. Both approaches mentioned in this paper achieve good performance on four widely adopted MuJoCo environments.
4. This paper demonstrates through an ablation experiment that the best results are achieved when both the actors and critics simultaneously engage in mutual imitation.

**Weaknesses:**

1. This paper does not provide the implementation code, which may limit the reproducibility of the experimental results. To improve transparency and reproducibility, it is recommended that the authors make the code publicly available, for instance by sharing a link to a GitHub repository, so that reviewers and readers can verify and reproduce the experiment.
2. This paper does not consider the weight between the Q function and the collaborative loss when computing the actor's gradient. It is recommended that the authors consider the weight, and conduct an ablation study on this parameter to explore the impact of different values on the performance in the test environments, or please give a justification for this fixed weight.

**Questions:**

1. Could the authors provide a more detailed explanation of how the mutual learning mechanism between actors directly impacts the exploration process and policy convergence? Specifically, how does minimizing the action differences between actors quantitatively lead to improved exploration outcomes?
2. In the selective imitation method, why was the value function chosen to determine when imitation should occur?
3. Have you considered adding a weight parameter between the Q function and the collaborative loss when computing the actor's gradient? Additionally, how does this hyperparameter influence the experimental performance?

---

### Official Review · Reviewer_UhVU · 2024-11-03

**Soundness:** 2
**Presentation:** 2
**Contribution:** 2
**Rating:** 3
**Confidence:** 3

**Summary:**

The paper presents a training-method to improve double actor reinforcement learning. They propose to use an additional loss that minimizes the discrepancy between actors and ciritic predictions during training. The authors present two variants of this "collaborative loss". The authors show how this loss can be incorporated into the implementation of a wide range of existing double-actor and multi-critic RL algorithms. They highlight benchmark results for four Mujoco environments, and show the developed approach matches or outperforms existing RL algorithms without the collaborative loss.

**Strengths:**

Strengths:
- The overall writing is fine; I could follow the paper well
- The method is simple and, indeed, seems widely applicable to many existing algorithms
- The method is described in reasonable detail, which allows to understand the methodology well
- Many baseline algorithms are used for comparison

**Weaknesses:**

Weaknesses:
- There seems to exist quite a bit of work that uses some sort of interaction/regularization between multiple actors (e.g., Li2023 et al.: Keep Various Trajectories: Promoting Exploration of Ensemble Policies in Continuous Control), indeed going beyond two actor-critic pairs, so I do not feel the statement “Existing double-actor DRL methods involve each actor-critic pair exploring independently” is warranted
- Given the close results compared to baseline algorithms, the limited choice of environments (just 4 Mujoco environments) is a bit lacking; showcasing environments with weak performance of other methods, and highlighting the performance the proposed method would have been more convincing
- Some of the reported baseline benchmark results seem off, e.g., CAL reports > 5000 reward for Ant in their implementation at 1M steps (Figure 2, Li et al., Simultaneous Double Q-learning…), which deviates very significantly from the reported reward. In Table 1, for SD3, 1750 is reported for Hopper; however, for Hopper in the SD3 paper, it seems to be much closer to 3500 than 2882; similarly >4000 for Ant is reported in the SD3 paper compared to the score of 1176 given in this paper. While I accept that there can be differences based on random seeds and unreported hyperparameter settings, these deviations seem pretty large. Can the authors please check these numbers or explain the discrepancies?
- Overall, I do not find the empirical results to be compelling; I do not see that the proposed approach significantly outperforms the existing methods

Minor:
- For intrinsic exploration, there should be more foundational related work, which should be cited
- Sometimes, the terminology in the related work is weak, e.g., “Current mainstream DRL algorithms typically utilize an architecture based on one actor and two critics.” -> That is the case for some algorithms like SAC or Double-DQN, but not necessarily for others; would can be more specific here
- Section 4.4. adds relatively little new information; I think it’s pretty straightforward, and a statement like “space complexity increases due to the parameters of the value network, which includes a five-layer Multilayer Perceptron” does seem too specific to the given implementation
- The term “mutual information” is an established term, and it is not obvious that the difference between actors/critics (4.2. (2), (4)) reduces mutual information
- Section 4.5 is difficult to follow in the main paper; maybe this can be re-structured not entirely to rely on the appendix

**Questions:**

Clarification:
- The “collaborative loss” tries to minimise the difference between policies and q-function estimates (compared to existing work which generally advocates for improved diversity). I feel that, in effect, the algorithm should behave more similar to a normal “single-model” algorithm, like TD3. Doesn’t this go against the spirit of double actor-learning?
- 4.2 (2) also indicates that the critic are optimised to predict the same target value Q-hat, which means that the predictions are even more similar. Can you provide some plot or analysis that showcases how different policy predictions and critic predictions actually are? I would really like to see a comparison to non double-actor methods.

---

### Official Review · Reviewer_7ket · 2024-11-05

**Soundness:** 3
**Presentation:** 3
**Contribution:** 3
**Rating:** 5
**Confidence:** 4

**Summary:**

This paper addresses two significant challenges in Deep Reinforcement Learning (DRL): enhancing exploration capabilities and improving the accuracy of Q-value estimation. The authors observe that existing double-actor DRL methods, while promising, suffer from a lack of cooperation between the two actors, leading to suboptimal policies. To mitigate this, they propose a competitive-cooperative actor-critic framework that encourages mutual learning among actors by minimizing the differences in their output actions and the discrepancies in the Q-values estimated by their respective critics. They present two specific implementations of their method and extend it to multi-critic DRL methods. The effectiveness of their approach is demonstrated through extensive experiments on four MuJoCo tasks, where it enhances the performance of nine state-of-the-art DRL methods in terms of return.

**Strengths:**

1.	Generality and Flexibility: The proposed framework is designed to be generic and can be seamlessly integrated into existing double-actor DRL methods as well as extended to multi-critic DRL methods. This broad applicability enhances its potential impact on the DRL community.
2.	Improved Policy Performance: By promoting mutual imitation among actors, the framework addresses the issue of independent exploration in existing methods, leading to the development of better policies and improved performance.
3.	Concrete Implementations: The paper provides two specific implementations of the framework, demonstrating its flexibility and practical applicability.
4.	Comprehensive Experiments: The authors conduct extensive experiments on four MuJoCo tasks and evaluate their method against nine state-of-the-art DRL algorithms. The results show consistent performance improvements, supporting the effectiveness of their approach.

**Weaknesses:**

1.	Experimental Scope: The experiments are restricted to MuJoCo tasks. Expanding validation to environments with more complex and dynamic variables, or to real-world tasks, would provide stronger evidence for the framework’s generalizability and practical relevance.
2.	Limited Analysis of Implementation Trade-offs: While both implementations of collaborative loss show performance improvements, the paper could benefit from a more nuanced discussion on the trade-offs between complexity and performance across various use cases.

**Questions:**

1.	Extension to Complex Tasks: Could the authors elaborate on the applicability of the competitive-cooperative framework in more complex, non-simulation tasks?
2.	Trade-offs in Collaborative Loss Implementations: What specific use cases would benefit more from each implementation of the collaborative loss? Are there trade-offs between performance and computational complexity that should be considered?
3.	Exploration and Exploitation Balance: How does the framework affect the exploration-exploitation balance compared to other multi-actor DRL methods, and how would this influence application to environments with sparse rewards?
4.	Impact of Q-value Discrepancy Minimization: Given the role of Q-value minimization in aligning actor policies, are there specific scenarios or tasks where this could inadvertently limit exploration or policy diversity?

---

### Note · Authors · 2024-11-21

I have read and agree with the venue's withdrawal policy on behalf of myself and my co-authors.